

# Non-invertible symmetries in finite-group gauge theory

Clay Córdova[1], Davi Bastos Costa[1] and Po-Shen Hsin[2,3]

**1** Enrico Fermi Institute & Kadanoff Center for Theoretical Physics, University of Chicago, USA
**2** Mani L. Bhaumik Institute for Theoretical Physics, Department of Physics and Astronomy, University of California, Los Angeles, CA 90095, USA
**3** Department of Mathematics, King's College London, Strand, London WC2R 2LS, UK

## Abstract

We investigate the invertible and non-invertible symmetries of topological finite-group gauge theories in general spacetime dimensions, where the gauge group can be abelian or non-abelian. We focus in particular on the 0-form symmetry. The gapped domain walls that generate these symmetries are specified by boundary conditions for the gauge fields on either side of the wall. We investigate the fusion rules of these symmetries and their action on other topological defects including the Wilson lines, magnetic fluxes, and gapped boundaries. We illustrate these constructions with various novel examples, including non-invertible electric-magnetic duality symmetry in 3+1d $\mathbb{Z}_2$ gauge theory, and non-invertible analogs of electric-magnetic duality symmetry in non-abelian finite-group gauge theories. In particular, we discover topological domain walls that obey Fibonacci fusion rules in 2+1d gauge theory with dihedral gauge group of order 8. We also generalize the Cheshire string defect to analogous defects of general codimensions and gauge groups and show that they form a closed fusion algebra.

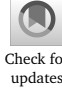

# 1 Introduction

Symmetries are powerful tools for understanding quantum systems. For instance, symmetries can provide hints about the long-distance behavior of physical systems even when they become strongly coupled. A famous example is the Lieb-Schultz-Mattis (LSM) theorem that constrains the dynamics of lattice models and the connection of these ideas to 't Hooft anomalies of generalized symmetries [1–6].

The notion of symmetry was given an intrinsic definition in terms of topological operators and their correlation functions in [7]. A frontier area of exploration is symmetries in field theories in general spacetime dimensions, where the topological operators or the corresponding topological quantum field theories are described by higher fusion categories.[1] Topological quantum field theories in general spacetime dimensions also play important role in exploring the dynamical consequences of symmetry in general gapped or gapless quantum systems such as constraining whether the symmetry can be realized on the boundary by symmetric gapped phases or trivially gapped phases [11–15] (see also [16] for related constructions). Thus, understanding the properties of topological operators or topological quantum field theories in general spacetime dimensions is important for learning dynamics of general quantum systems from symmetry.

Finite-group topological gauge theories [17] provide a fruitful playground to explore these notions. They can arise in various settings such as gapped phases of lattice models and quantum field theories and topological codes. Finite-group gauge theories are also relevant in experiments, most prominently $\mathbb{Z}_2$ gauge theory in s-wave superconductors where $U(1)$ electromagnetism is broken to $\mathbb{Z}_2$ by Cooper pairing [18]. In recent years, there are also experimental realizations of ground state wavefunctions for gauge theories with $\mathbb{Z}_2$ gauge group (e.g., [19]) and dihedral gauge group of order 8 [20] in 2+1d by quantum processors.

---

[1]In this work we will only consider fully topological operators, leaving cases with general subsystem symmetries to future work. See e.g., [8–10] and the references therein for examples of gapped domain walls and interfaces in fracton models.

The invertible symmetries, i.e., symmetries that have an inverse transformation, in finite-group gauge theories have been discussed extensively in the literature of symmetry-enriched topological orders (SET) [21–26]. In particular, recently invertible symmetries in finite-group gauge theories have important applications to new fault-tolerant logical gates in topological quantum codes [27–34]. Invertible symmetries in finite-group gauge theories also provide construction for new automorphism codes, e.g., [35–39].

In addition to invertible symmetries, finite-group gauge theories can have non-invertible symmetry, where the generating topological operators do not obey group-law fusion, and in particular do not have an inverse. Such non-invertible topological defects can be present in various gapless or gapped quantum systems, see [40–49] for early work on this subject. Meanwhile, examples of non-invertible topological domain wall defects in finite-group gauge theory are discussed in various literature [23, 28, 50–59]. In 2+1d, such topological domain walls or boundaries correspond to certain condensation of bulk topological excitations called Lagrangian algebras [60]. On the other hand, the gapped domain walls and boundaries in higher dimensions are less understood (see [61–63] for recent studies for the gapped boundaries of $\mathbb{Z}_2$ gauge theory in 3+1d).

In this work, we will investigate general symmetries, both invertible and non-invertible, in finite-group topological gauge theories. We will focus on the topological domain walls in general spacetime dimension. Since topological finite-group gauge theories are naturally defined on the lattice (see e.g., [17]), we will investigate the symmetries by placing the theories on the lattice. A companion paper [100] will explore the relationship of these symmetries to condensations.

## 1.1 Summary of results

Gauge theories with a finite gauge group $G$ can be defined by a path integral on the lattice [17]. A flat gauge field configuration is a map that assigns to each oriented edge a group element $g_{ij} \in G$ and satisfies a flatness condition for every 2-simplex of the triangulated manifold. Gauge transformations are maps that assign to each vertex a group element $h_i \in G$ and transform a flat gauge field configuration $g_{ij}$ to $h_i g_{ij} h_j^{-1}$. The total action is a product of local terms classified by group cohomology $H^D(G, U(1))$ whose elements are functions that assign a well-defined phase depending on the values of the gauge field configuration in each $D$-simplex. The partition function is then given by a summation over gauge equivalence classes of flat gauge field configurations and is weighted by the topological action (see Section 2 for a review).

**Domain walls and gapped boundaries on the lattice** Gapped boundaries of untwisted gauge theory with a finite gauge group $G$ in general dimension can be constructed from subgroups $K \leq G$ and a choice of topological action $\alpha \in H^{D-1}(K, U(1))$. Given this data, we construct a gapped boundary $\mathcal{B}_{K,\alpha}$ by restricting the gauge field configurations to be in the subgroup $K$ and by attaching the topological action $\alpha \in H^{D-1}(K, U(1))$ along the boundary $\partial \mathcal{M}$. Motivated by the folding trick, we construct a domain wall $\mathcal{D}_{H,\alpha}(\Sigma)$ by having gauge fields on the subgroup $H \leq G \times G$ and by attaching the topological action $\alpha \in H^{D-1}(H, U(1))$ along the codimension-one submanifold $\Sigma$.

**Fusion of domain walls and action on gapped boundaries** Despite being simple, this definition is generic because it applies to any group $G$ and dimension $D$. Furthermore, some of the fusion rules for the codimension-one topological operators can be derived in a very simple way from this description. One of the main contributions of this paper is to derive the fusion ring structure of the domain walls $\mathcal{D}_{H,\alpha}$ with subgroup $H \leq G \times G$ and topological action $\alpha \in H^{D-1}(H, U(1))$ as elements of one of the following two families:

- $H = K^{(\phi)} \equiv \{(\phi \cdot k, k) : k \in K\}$ with $K \lhd G$ and $\phi \in \text{Aut}(G)$ and a topological action $\alpha \in H^{D-1}(K, U(1))$ evaluated on the right entry of $K^{(\phi)}$.

- $H = K_L \times K_R$, with $K_L, K_R \lhd G$ and $\alpha = \alpha_L \times \alpha_R$ with $\alpha_L \in H^{D-1}(K_L, U(1))$ and $\alpha_R \in H^{D-1}(K_R, U(1))$.

We show that these two families are generated by the domain walls:

- Automorphism domain walls: $\mathcal{D}_{G^{(\phi)}}$, with $\phi \in \text{Aut}(G)$;

- Diagonal domain walls: $\mathcal{D}_{K^{(\text{id})}, \alpha}$, with $K \lhd G$ and $\alpha \in H^{D-1}(K, U(1))$;

- Magnetic domain wall: $\mathcal{D}_{G \times G}$;

which obey the following fusion rules:

$$\mathcal{D}_{G^{(\phi)}} \times \mathcal{D}_{G^{(\phi')}} = \mathcal{D}_{G^{(\phi \circ \phi')}}, \tag{1.1}$$

$$\mathcal{D}_{K^{(\text{id})}, \alpha} \times \mathcal{D}_{K'^{(\text{id})}, \alpha'} = \frac{|G|}{|K \cdot K'|} \mathcal{D}_{(K \cap K')^{(\text{id})}, \alpha \cdot \alpha'}, \tag{1.2}$$

$$\mathcal{D}_{K_L^{(\text{id})}, \alpha_L} \times \mathcal{D}_{G \times G} \times \mathcal{D}_{K_R^{(\text{id})}, \alpha_R} = \mathcal{D}_{K_L \times K_R, \alpha_L \times \alpha_R}, \tag{1.3}$$

$$\mathcal{D}_{G \times G} \times \mathcal{D}_{K^{(\text{id})}, \alpha} \times \mathcal{D}_{G \times G} = \mathcal{Z}(K, \alpha) \mathcal{D}_{G \times G}, \tag{1.4}$$

$$\mathcal{D}_{G^{(\phi)}} \times \mathcal{D}_{K^{(\text{id})}, \alpha} = \mathcal{D}_{K^{(\phi)}, \alpha}, \tag{1.5}$$

$$\mathcal{D}_{K^{(\text{id})}, \alpha} \times \mathcal{D}_{G^{(\phi)}} = \mathcal{D}_{G^{(\phi)}} \times \mathcal{D}_{\phi^{-1}(K), \phi^* \alpha} = \mathcal{D}_{(\phi^{-1}(K))^{(\phi)}, \phi^* \alpha}, \tag{1.6}$$

$$\mathcal{D}_{G^{(\phi)}} \times \mathcal{D}_{G \times G} = \mathcal{D}_{G \times G} \times \mathcal{D}_{G^{(\phi)}} = \mathcal{D}_{G \times G}, \tag{1.7}$$

with $\phi \circ \phi'$ the automorphism composition of $\phi, \phi' \in \text{Aut}(G)$; $\alpha \cdot \alpha'|_{K \cap K'} \in H^{D-1}(K \cap K', U(1))$; $|G|/|K \cdot K'|$ the 0-form partition function of $G/K \cdot K'$ gauge theory on $\Sigma$; $\mathcal{Z}(K, \alpha)$ the partition function of $K$ gauge theory twisted by $\alpha$ on $\Sigma$; and $\phi^* \alpha$ the pullback of $\alpha$ by $\phi : \phi^{-1}(K) \to K$.

In addition, we show that the domain walls that generate this fusion ring have the following action on the gapped boundaries:

$$\mathcal{D}_{G^{(\phi)}} \times \mathcal{B}_{K, \alpha} = \mathcal{B}_{\phi(K), \phi^{-1*} \alpha}, \tag{1.8}$$

$$\mathcal{D}_{K^{(\text{id})}, \alpha} \times \mathcal{B}_{K', \alpha'} = \frac{|G|}{|K \cdot K'|} \mathcal{B}_{K \cap K', \alpha \cdot \alpha'}, \tag{1.9}$$

$$\mathcal{D}_{G \times G} \times \mathcal{B}_{K, \alpha} = \mathcal{Z}(K, \alpha) \mathcal{B}_G. \tag{1.10}$$

**Transformation on other operators**   Group elements and gauge transformations along $\Sigma$ in the presence of $\mathcal{D}_{H, \alpha}(\Sigma)$ are restricted to the subgroup $H$. From this feature, we can derive the transformation of other operators on the domain walls. As an example, it is easy to show that:

$$\mathcal{D}_1 \cdot W_{\rho_i} = \sum_{\rho_k \in \text{irreps}} d_i d_k W_{\rho_k}, \qquad \mathcal{D}_1 \cdot M_g = 0, \tag{1.11}$$

$$\mathcal{D}_{G \times G} \cdot W_{\rho_i} = 0, \qquad \mathcal{D}_{G \times G} \cdot M_g = \sum_{[k] \in \text{Cl}(G)} M_k, \tag{1.12}$$

$$\mathcal{D}_{G^{(\phi)}} \cdot W_{\rho_i} = W_{\rho_i \cdot \phi^{-1}}, \qquad \mathcal{D}_{G^{(\phi)}} \cdot M_g = M_{\phi(g)}. \tag{1.13}$$

for all simple Wilson lines $W_{\rho_i}$ and magnetic defects $M_g$ where $d_i$ is the dimension of the irreducible representation $\rho_i$.

**Higher codimensional topological operators: Cheshire strings**  In the definition of the domain wall $\mathcal{D}_{H,\alpha}(\Sigma)$, a crucial ingredient is the orientation of the normal bundle $N\Sigma$. It allows us to consistently define the global meaning of left and right associated with the left and right components of the subgroup $H < G \times G$. Diagonal domain walls, however, are orientation reversal invariant and can be generalized as higher codimensional operators. The dimension-$n$ generalization of the diagonal domain walls are classified by subgroups $K < G$ and a topological action $\alpha \in H^n(K, U(1))$ and obey the fusion rule:

$$\mathcal{D}_{K^{(\mathrm{id})},\alpha}(\Sigma_n) \times \mathcal{D}_{K'^{(\mathrm{id})},\alpha'}(\Sigma_n) = \frac{|G|}{|K \cdot K'|} \mathcal{D}_{(K \cap K')^{(\mathrm{id})}, \alpha \cdot \alpha'}(\Sigma_n), \qquad (1.14)$$

with $\Sigma_n$ a $n$-dimensional submanifold of $\mathcal{M}$. This fusion rule generalizes the fusion rule of Cheshire strings [64, 65].

**Non-invertible electric-magnetic duality domain wall**  Note that in the data that specifies a domain wall, dimension dependence comes from the topological action $\alpha \in H^{D-1}(H, U(1))$. By working out the particular case of $G = \mathbb{Z}_2$ gauge theory in $D = 3$, we compute the fusion, action on boundaries and transformation of other operators for the domain wall associated with the subgroup $H = \mathbb{Z}_2 \times \mathbb{Z}_2 \lhd \mathbb{Z}_2 \times \mathbb{Z}_2$ with the non-trivial topological action $\alpha_2 \in H^2(H, U(1)) = \mathbb{Z}_2$. We find

$$\mathcal{D}_{\mathbb{Z}_2 \times \mathbb{Z}_2, \alpha_2} \times \mathcal{D}_{\mathbb{Z}_2 \times \mathbb{Z}_2, \alpha_2} = 1\,, \qquad (1.15)$$

$$\mathcal{D}_{\mathbb{Z}_2 \times \mathbb{Z}_2, \alpha_2} \times \mathcal{B}_1 = \mathcal{B}_{\mathbb{Z}_2}\,, \qquad\qquad \mathcal{D}_{\mathbb{Z}_2 \times \mathbb{Z}_2, \alpha_2} \times \mathcal{B}_{\mathbb{Z}_2} = \mathcal{B}_1\,, \qquad (1.16)$$

$$\mathcal{D}_{\mathbb{Z}_2 \times \mathbb{Z}_2, \alpha_2} \cdot W = M\,, \qquad\qquad \mathcal{D}_{\mathbb{Z}_2 \times \mathbb{Z}_2, \alpha_2} \cdot M = W\,, \qquad (1.17)$$

showing that $\mathcal{D}_{\mathbb{Z}_2 \times \mathbb{Z}_2, \alpha_2}$ is the electric-magnetic duality symmetry defect. The procedure we follow for the computation is more general and shows that $\mathcal{D}_{G \times G, \alpha}$ generalizes the electric-magnetic duality to higher dimensions, generic gauge groups $G$ and topological action $\alpha \in H^{D-1}(G, U(1))$. This class of domain walls mixes invertible electric and magnetic operators and obeys a non-invertible fusion in general. For instance, in the theory with $G = \mathbb{D}_4$ (the dihedral group of order 8), and $D = 3$, the domain wall associated with the subgroup $H = \mathbb{D}_4 \times \mathbb{D}_4$ and the non-factorized element $\alpha_2 \in H^2(\mathbb{D}_4 \times \mathbb{D}_4, U(1)) = \mathbb{Z}_2 \times \mathbb{Z}_2$, obeys the Fibonacci fusion rule:

$$\mathcal{D}_{\mathbb{D}_4 \times \mathbb{D}_4, (1,1)} \times \mathcal{D}_{\mathbb{D}_4 \times \mathbb{D}_4, (1,1)} = 1 + \mathcal{D}_{\mathbb{D}_4 \times \mathbb{D}_4, (1,1)}\,, \qquad (1.18)$$

and mixes magnetic and electric operators.

## 2  Review of finite-group gauge theory on the lattice

In this section we will review basic properties of finite-group gauge theories on the lattice. These theories can be defined in general spacetime dimension $D$ and are classified by tuples $(G, [\alpha_D])$ with $G$ a finite group and $[\alpha_D] \in H^D(G, U(1))$ a $D$-cohomology class. Such theories can describe liquid gapped phases, i.e., gapped phases with fully mobile excitations. While gapped phases in $D = 3$ can be described by modular tensor category, gapped phases in $D = 4$ and higher spacetime dimension are more constrained, and topological finite-group gauge theories provide an important class of representative examples [66–68]. Furthermore, these theories are examples of topological gauge theories and provide an elementary illustration of the categorical approach to quantum field theory [69, 70]. Now we summarize a few of its properties.

- **Gauge field configurations and gauge transformations.** Let us denote the gauge group by $G$, and the spacetime manifold by $\mathcal{M}$ of general spacetime dimension $D \geq 2$. We assume $\mathcal{M}$ is orientable and connected. We triangulate the spacetime manifold, enumerate its vertices $\{v_i : 0 \leq i \leq n\}$, and define a *gauge field configuration* as a map $\vec{g}$ that assigns to each edge $[v_i, v_j]$ such that $i < j$ a group element $g_{ij} \equiv \vec{g}([v_i, v_j]) \in G$. A *path* in the triangulation of $\mathcal{M}$ is a sequence of vertices connected by edges $\gamma = (v_{i_1}, \ldots, v_{i_n})$, and the *holonomy* along a closed path (with $i_1 = i_n$) is defined by

$$g_\gamma = g_{i_1 i_2} \cdots g_{i_{n-1} i_n} , \tag{2.1}$$

where $g_{ij} \equiv g_{ji}^{-1}$ whenever $i > j$. A gauge field configuration is said to be *flat* if the holonomy (flux) along the boundary of every 2-simplex $[v_i, v_j, v_k]$ of the triangulation of $\mathcal{M}$ is trivial:

$$g_{(v_i, v_j, v_k)} = g_{ij} \cdot g_{jk} \cdot g_{ki} = 1 . \tag{2.2}$$

This local flatness condition implies that the holonomy along a closed loop depends only on the homotopy class of the path $\gamma \in \pi_1(\mathcal{M})$. Therefore, a *flat gauge field configuration* can be described globally by a *flat connection* $a \in \mathrm{Hom}(\pi_1(\mathcal{M}), G)$ where $g_\gamma = a(\gamma)$, and Hom indicates that $a$ defines a group homomorphism under concatenation of loops in $\mathcal{M}$.

The gauge field configurations $\vec{g}$ and $\vec{g}\,'$ are *gauge equivalent* if

$$g_{ij}' = h_i \cdot g_{ij} \cdot h_j^{-1} , \tag{2.3}$$

for some map $\vec{h}$ that assigns to each vertex $v_i$ a group element $h_i \equiv \vec{h}(v_i) \in G$. We call the map $\vec{h}$ a *gauge transformation* and we say that it changes the gauge field configuration from $\vec{g}$ to $\vec{g}\,'$. Conversely, two flat connections $a, a' \in \mathrm{Hom}(\pi_1(\mathcal{M}), G)$ are *gauge equivalent* if there exists $h \in G$ such that $a'(\gamma) = h \cdot a(\gamma) \cdot h^{-1}$ for every $\gamma \in \pi_1(\mathcal{M})$. We denote this set by $\mathrm{Hom}(\pi_1(\mathcal{M}), G)/G$.

- **Topological action and group cohomology.** The total action is a product of local terms, one for each $D$-simplex of the triangulation of $\mathcal{M}$ (which we also denote by $\mathcal{M}$), and is given by

$$\prod_{[v_{i_1}, \ldots, v_{i_{D+1}}] \in \mathcal{M}} \alpha_D(g_{i_1 i_2} \ldots, g_{i_D i_{D+1}})^{\epsilon_i} , \tag{2.4}$$

with $\epsilon_i = \pm 1$ depending on whether the orientation of the $D$-simplex agrees with that of $\mathcal{M}$ and with $[\alpha_D] \in H^D(G, U(1))$.[2] The $n$-th *group cohomology* $H^n(G, U(1))$ is a finite abelian group defined as the quotient of $n$-cocycles by $n$-coboundaries. Specifically, the set of $n$-cochains $C^n$ is the set of functions $\alpha_n : G^n \to U(1)$ and the coboundary operator $\delta^{(n)} : C^n \to C^{n+1}$ is

$$\delta^{(n)}\alpha_n(g_1, \ldots, g_{n+1}) = \alpha_n(g_1, \ldots, g_n)^{(-1)^{n+1}} \alpha_n(g_2, \ldots, g_{n+1})$$
$$\times \prod_{i=1}^{n} \alpha_n(g_1, \ldots, g_i \cdot g_{i+1}, \ldots, g_{n+1})^{(-1)^i} . \tag{2.5}$$

The set of $n$-cocycles is defined by $Z^n(G, U(1)) = \{\alpha_n \in C^n : \delta^{(n)}\alpha_n = 1\}$ and the set of $n$-coboundaries by $B^n(G, U(1)) = \{\alpha_n \in C^n : \alpha_n = \delta^{n-1}\alpha_{n-1}, \text{ with } \alpha_{n-1} \in C^{n-1}\}$. It follows from the definition of the coboundary operator that $\delta^{(n)} \cdot \delta^{(n-1)} = 1$ so that the

---

[2]The positive orientation of the $D$-simplex is obtained by having $i_1 < \cdots < i_{D+1}$.

set of $n$-coboundaries is a subgroup of the set of $n$-cocycles. The $n$-th group cohomology of algebraic cocycles of $G$ with $U(1)$ coefficients is defined by:

$$H^n(G, U(1)) = Z^n(G, U(1))/B^n(G, U(1)) = \text{Ker } \delta^{(n)}/\text{Im } \delta^{(n-1)}. \tag{2.6}$$

The fact that the topological action does not depend on the choice of triangulation of $\mathcal{M}$ follows from the cocycle condition $\delta^{(D)}\alpha_D = 1$. When no confusion is possible we will drop the subscript $n$ in $\alpha_n$ and for convenience, we are going to denote by $\alpha_n$ the $n$-cohomology class and the cocycle used to represent it.

- **Partition function.** We denote by $\mathcal{Z}(G, \mathcal{M}, \alpha_D)$ the gauge theory partition function associated with the finite-group $G$ and local action $\alpha_D \in H^D(G, U(1))$ on $\mathcal{M}$. We say the theory is *untwisted* if $\alpha_D$ is trivial and *twisted* otherwise. In the first case, we suppress the symbol for the local action. The partition function $\mathcal{Z}(G, \mathcal{M}, \alpha_D)$ on the lattice is given by a summation over gauge equivalence classes (2.3) of flat gauge field configurations (2.2) weighted by the topological action (2.4) and normalized by $1/|G|$. This local lattice definition can be recast in a global and manifestly topological invariant way as

$$\mathcal{Z}(G, \mathcal{M}, \alpha_D) = \frac{1}{|G|} \sum_{a \in \text{Hom}(\pi_1(\mathcal{M}), G)/G} \langle a^* \alpha_D, [\mathcal{M}] \rangle, \tag{2.7}$$

with $[\mathcal{M}]$ the fundamental class of $\mathcal{M}$ and $\alpha_D \in H^D(BG, U(1))$. Above we used the fact that there is a isomorphism between group cohomology $H^D(G, U(1))$ and topological co-homology $H^D(BG, U(1))$ where $BG$ is a classifying space for $G$ (a space with $\pi_1(BG) = G$ and $\pi_n(BG) = 1$ for $n > 1$). In this setup, the summation is over principal $G$ bundles over $\mathcal{M}$ and the flat connection $a$ defines a homotopy class of maps $a : \mathcal{M} \to BG$ which we use to pull back $\alpha_D$ to spacetime. We see that the theory can be viewed as a sigma model with target space the classifying space $BG$ [71].

The normalization factor $1/|G|$ is such that the partition function for untwisted $G$ gauge theory on $S^1 \times S^{D-1}$ equals

$$\mathcal{Z}(G, S^1 \times S^{D-1}) = \begin{cases} 1, & D \geq 3, \\ |G|, & D = 2, \end{cases} \tag{2.8}$$

which is the dimension of the Hilbert space on $S^{D-1}$. For $D \geq 3$, the dimension is always one since $S^{D-1}$ is simply connected. For $D = 2$, the space is a circle, and the dimension of Hilbert space is $|G|$. (Recall that we suppress the symbol for the topological action when it is trivial.)

When $G$ is a finite abelian group the theory can be generalized to higher-form $G$ gauge theory. In a $p$-form $G$ gauge theory, the gauge field configurations are maps that assign group elements to $p$-simplices, the flatness condition involves the boundary of $(p+1)$-simplices, gauge transformations come from $(p-1)$-simplices and the topological action is classified by $H^D(B^p G, U(1))$ ($B^p G$ is a space with $\pi_p(B^p G) = G$ and $\pi_n(B^p G) = 0$ otherwise). The partition function for untwisted $p$-form $G$ gauge theory is proportional to $|H^p(\mathcal{M}, G)|$ which equals (2.7) for $p = 1$. This generalization does not work for non-abelian $G$ because of the flatness condition except the $p = 0$ case. When $p = 0$, a gauge field configuration is a map $\vec{g}$ that assigns a gauge group element to every vertex of $\mathcal{M}$, $\vec{g}(v_i) \equiv g_i$. By the flatness condition $g_{[v_i, v_j]} = g_i g_j^{-1} = 1$ for all edges of $\mathcal{M}$. One finds that the map $\vec{g}$ assigns the same group element to every connected component of $\mathcal{M}$ and therefore

$$\mathcal{Z}^0(G, \mathcal{M}) = |G|^{|\pi_0(\mathcal{M})|}. \tag{2.9}$$

Below we often assume that the spacetime manifold is connected in which case the above is simply $\mathcal{Z}^0(G, \mathcal{M}) = |G|$.

- **Hilbert space.** Consider canonical quantization on $\mathcal{M} = \mathbb{R}_{\text{time}} \times M_{\text{space}}$. The partition function on $S^1 \times M_{\text{space}}$ gives the dimension of the Hilbert space on $M_{\text{space}}$ and can be computed explicitly using the lattice definition. If we view the partition function as a summation over flat connections as in equation (2.7) then, for a gauge field with value $g$ in the time direction, the field configurations on $M_{\text{space}}$ that label the physical Hilbert space on $M_{\text{space}}$ correspond to the flat connections such that the compactification of the topological action $\alpha_D \in H^D(BG, U(1))$ on $S^1$ is trivial:

$$a^\star i_g \alpha_D = 0 \mod 2\pi\mathbb{Z}, \quad \forall g \in G, \tag{2.10}$$

where $i_g \alpha$ is the slant product (see e.g., Appendix A of [72]).[3] The condition (2.10) can also be viewed as the "equation of motion" for the field variation in the temporal direction by the amount $g$.

In the case of vanishing $\alpha_D$ this Hilbert space is spanned by basis vectors in one-to-one correspondence with elements of $\text{Hom}(\pi_1(M_{\text{space}}), G)/G$ where the quotient is the action by $G$ conjugation, see (2.3).

- **Wilson lines.** Wilson lines are one-dimensional extended operators labeled by representations of the gauge group. The Wilson line associated with the representation $\rho$ inserted on a loop $\gamma$ is given by

$$W_\rho(\gamma) = \chi_\rho(g_\gamma) = \text{Tr}_\rho(g_\gamma), \tag{2.11}$$

where $\chi_\rho : G \to \mathbb{C}$ is the character (trace) of the representation $\rho$ and $g_\gamma \in G$ is the holonomy around $\gamma$. The operator $W_\rho(\gamma)$ depends on the homotopy class of the cycle $\gamma$. We recall that the fundamental group depends on a choice of basepoint. Assuming that the spacetime manifold is connected nothing depends on this choice. However, the presence of a basepoint implies that a loop $\gamma$ homotopic to $\gamma_1 \cdot \gamma_2$ cannot be viewed as the disjoint union of $\gamma_1$ and $\gamma_2$. Therefore, in general $W_\rho(\gamma) \neq W_\rho(\gamma_1)W_\rho(\gamma_2)$, even if $\gamma_1 \cdot \gamma_2$ is homotopic to $\gamma$. An important exception is when $\rho$ is one-dimensional, which is the case for all irreducible representations of abelian groups. We say that a Wilson line $W_\rho$ has electric charge $\rho$.

If two Wilson lines are placed along the same loop they fuse according to the tensor product of representations. This follows from the fact that $\chi_\rho(g)\chi_{\rho'}(g) = \chi_{\rho \otimes \rho'}(g)$. Furthermore, representations of $G$ are spanned by irreducible representations. Therefore, given the Wilson lines in representation $\rho$ and $\rho'$ we have:

$$W_\rho(\gamma)W_{\rho'}(\gamma) = W_{\rho \otimes \rho'}(\gamma) = \sum_{\rho_i \in \text{irreps}} c_i W_{\rho_i}(\gamma), \tag{2.12}$$

with $c_i \in \mathbb{N}$ the coefficient of $\rho_i$ in the expansion of $\rho \otimes \rho'$ in irreducible representations.

- **General invertible electric defects.** General invertible electric defects are $n$-dimensional operators labeled by elements of $H^n(G, U(1))$, the $n$-th group cohomology of $G$ with

---

[3]Explicitly,

$$i_g \alpha_D(g_1, \cdots, g_{D-1}) = \alpha_D(g, g_1, \cdots, g_{D-1})^{(-1)^{D-1}} \alpha_D(g_1, \ldots, g_{D-1})$$

$$\times \prod_{j=1}^{D-2} \alpha_D(g_1, \cdots, g_j, (g_1 \cdot g_2, \cdots g_j)^{-1} \cdot g \cdot (g_1 \cdot g_2, \cdots g_j), \cdots, g_{D-1})^{(-1)^{D-1+j}}.$$

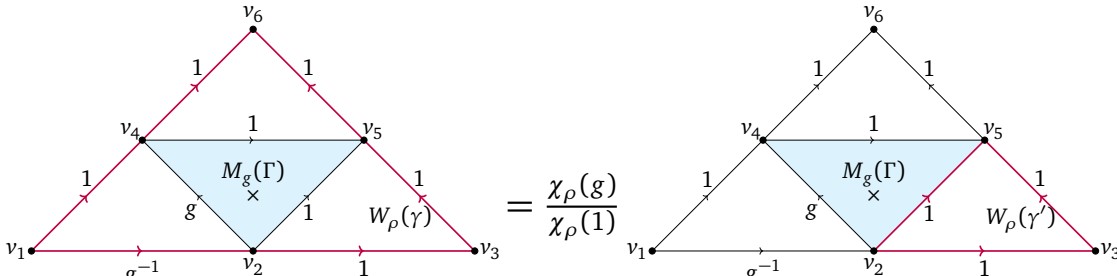

Figure 1: Example of valid gauge field configuration on the vicinity of the magnetic defect $M_g$ insertion (indicated by an ×), and its linking action on the Wilson line $W_\rho$ (shown in red). The magnetic insertion has linking number one with $\gamma = (v_2, v_4, v_5)$ associated with the 2-simplex $[v_2, v_4, v_5]$ so a valid gauge field configuration should satisfy $g_{(v_2, v_5, v_6)} = g$. Note, however, that the holonomy around the other 2-simplices is trivial. Furthermore, the expectation value for the two insertions with the Wilson line $W_\rho$ along $\gamma = (v_1, v_4, v_6, v_5, v_3, v_2, v_1)$ which is linked with $\Gamma$, and along $\gamma' = (v_2, v_5, v_3, v_2)$ which is unlinked with $\Gamma$ are related by $\frac{\chi_\rho(g)}{\chi_\rho(1)}$.

$U(1)$ coefficients (2.6). They are obtained by attaching a topological action along the $n$-dimensional manifold they are defined on. The general invertible electric operator associated with $\alpha_n \in H^n(G, U(1))$ inserted on the $n$-dimensional closed manifold $\Sigma_n$ is given by [28]:

$$W_{\alpha_n}(\Sigma_n) = \prod_{[v_{i_1}, \dots, v_{i_{n+1}}] \in \Sigma_n} \alpha_n(g_{i_1 i_2} \dots, g_{i_n i_{n+1}})^{\epsilon_i}. \tag{2.13}$$

For $n = 1$, we have $H^1(BG, U(1)) \cong \mathrm{Hom}(G, U(1))$ and these operators reduces to a Wilson line in a one-dimensional representation. For general $n$, they are submanifolds decorated with topological action for the $G$ gauge fields. Examples of these defects are studied in [27, 28, 53, 62, 64, 73].

If two general invertible electric defects are placed along the same $n$-dimensional closed submanifold $\Sigma_n$ they fuse according to the abelian group structure of $H^n(G, U(1))$. More precisely, given $\alpha_n, \alpha'_n \in H^n(G, U(1))$ we have:

$$W_{\alpha_n}(\Sigma_n) W_{\alpha'_n}(\Sigma_n) = W_{\alpha_n \cdot \alpha'_n}(\Sigma_n). \tag{2.14}$$

This is consistent with the property that fusing such domain walls is the same as first stacking the SPT phases with $G$ symmetry labeled by $\alpha_n, \alpha'_n$ on the wall and then gauging the $G$ symmetry [28].

- **Magnetic defects**. Magnetic defects are codimension-two operators labeled by conjugacy classes of $G$ [74, 75]. The insertion of a magnetic defect associated to the conjugacy class of some element $g \in G$ on a closed connected $(D-2)$-submanifold $\Gamma$ modifies the flatness condition (2.2) for the allowed gauge field configurations in the partition function. Specifically, for every 2-simplex $[v_i, v_j, v_k]$ such that $\gamma = (v_i, v_j, v_k)$ links with $\Gamma$ the insertion of $M_g(\Gamma)$ restricts the holonomy $g_\gamma$ to be $g$ instead of 1. Here, we view $\Gamma$ as being spanned by $(D-2)$-simplices in the dual triangulation of $\mathcal{M}$. Note that this implies that the Wilson line $W_\rho$ has nontrivial linking with magnetic defects $M_g$ given by $\frac{\chi_\rho(g)}{\chi_\rho(1)}$. In general, the operator $M_g(\Gamma)$ depends on the isotopy class of $\Gamma$. See Fig. 1 for illustration. We say that a magnetic defect $M_g$ has magnetic charge $g$.

- **Dyons.** In $D = 3$ magnetic defects are also one-dimensional, therefore one can consider more general one-dimensional operators with electric and magnetic charge. We will focus on the case $\alpha = 0$. This operators are called dyons and they are labeled by the tuple $([g], \rho)$ with $[g]$ a non-trivial conjugacy class of $G$ and $\rho$ a non-trivial representation of the group $C(g) = \{k \in G : kgk^{-1} = g\}$, the centralizer of a fixed element $g \in [g]$ (see e.g., [76]). Wilson lines are dyons with label $([1], \rho)$ since $C(1) \cong G$, and magnetic defects are dyons with label $([g], 1)$ with 1 the trivial representation of $C(g)$.

  In $D > 3$ with $\alpha = 0$, the magnetic defects have dimension $(D-2) > 1$. Since on the magnetic defect of conjugacy class $[g]$ the gauge group is reduced to the centralizer $C(g)$, an analog of a "dyon" defect can be defined by decorating the magnetic defect with an invertible electric defect for the unbroken gauge group $C(g)$, labelled by a $(D-2)$-cocycle $\beta \in H^{D-2}(C(g), U(1))$.

  When $\alpha \neq 0$, a general dyonic defect is given by decorating the magnetic defect with (higher) projective representation. See e.g., [28].

- **Fusion of magnetic defects in $D = 3$**. When the $G$ gauge theory has trivial topological action the magnetic defects obey the following fusion rules. Since the magnetic defect $M_g$ reduces the gauge group $G$ to the centralizer subgroup $C(g)$, the Wilson line in irreducible representation $\rho$ in the presence of the magnetic defect decomposes into $\sum_i W_{\rho_i}$ for $\rho = \bigoplus_i \rho_i$ under the stabilizer subgroup $C(g)$. Thus

$$W_\rho \times M_g = \sum_i W_{\rho_i} M_g, \tag{2.15}$$

  where the right-hand side above should be viewed as sum of dyons, and the sum over $i$ is as in the decomposition of $\rho$ above.

  Consider fusing magnetic defects $M_m, M_{\bar{m}}$. From the above discussion, this should give the condensation defect of Wilson lines that can terminate on the magnetic defect. Denote $\mathcal{R}_m$ to be the set of irreducible representations of $G$ whose decomposition under the stabilizer subgroup $C(m)$ contains the trivial representation. Then

$$M_m \times M_{\bar{m}} = \frac{1}{|G|} \sum_{g \in G} M_{mg\bar{m}g^{-1}} \sum_{r \in \mathcal{R}_m} d_r W_{\rho_r}, \tag{2.16}$$

  where $d_i$ is the dimension of the representation $\rho_i$, and the sum is over the representations in $\mathcal{R}_m$. On the right-hand side, we use the property that multiplying two conjugacy classes in general get multiple fusion outcomes. The coefficients on the right-hand side of (2.16) can be computed using the method in [77, 78]. Let us denote the right-hand side by $\mathcal{A}_m$, we want to find the coefficient of $W_{\rho_r}$ in $\mathcal{A}_m$. Consider fusing $\mathcal{A}_m$ with $\bar{W}_{\rho_r}$ on $S^{D-1} \times S^1$ with the lines wrapping $S^1$, the partition function computes $\text{Hom}(\mathcal{A}_m \times \bar{W}_{\rho_r}, 1)$ which is the desired coefficient. On the other hand, view $\mathcal{A}_m$ as an empty cylindrical tube, the configuration is topologically equivalent to $B^{D-1} \times S^1$ with punctured ball $B^{D-1}$ by Wilson line $W_{\rho_r}$ wrapping $S^1$, and thus the coefficient is $\dim \mathcal{H}(B^{D-1}, \rho_r)$, which equals to the dimension of the space of operators living at the intersection of the Wilson line and the boundary, i.e. the dimension of the representation.

  For example, if $m$ is in the center of $G$, the stabilizer $C(m) = G$ is the entire group, then $\mathcal{R}_m$ only contains the trivial representation. The fusion of the magnetic defects reduces to $M_m \times M_{m'} = M_{mm'}$.

  When $\alpha \neq 0$, magnetic defects carry additional projective representations, which modify the fusion rules as discussed in [28].

- **Hamiltonian formalism (quantum double model)**. We can also consider a Hamiltonian formalism with continuous time and discrete space. One possible Hamiltonian model is the quantum double (or its twisted version when the theory has a topological action) discussed in [50, 79, 80]. This theory should be viewed as an ultra-violet extension of topological finite-group gauge theory discussed above. Specifically, this Hamiltonian model has excitations with nonzero energy, and its Hilbert space is the tensor product of local Hilbert spaces $\mathbb{C}[G]$ on each edge with basis $\{|g\rangle : g \in G\}$. At low energy, with particular couplings, the ground states realize the Hilbert space of the topological finite-group gauge theory.

More concretely, the topological $G$ gauge theory is realized in the low energy ground states by imposing an energy cost for the configuration that violates the Gauss law $\nabla \cdot E = 0$ for electric field $E$. To realize the flatness condition on the gauge fields, we also need to impose an energy cost for the fluxes. Thus, the Hamiltonian has the form

$$H = -\sum A_v - \sum_f B_f \,, \tag{2.17}$$

where $A_v$ is the energy cost for violation of Gauss law at vertex $v$, and $B_f$ is the energy cost for fluxes on face $f$. The explicit form of $A_v, B_f$ are given in [50, 79, 80].

# 3 Lattice construction of topological operators

This section will discuss topological defects in finite-group gauge theory. We will focus on codimension-one topological defects. They correspond to an ordinary symmetry of the theory when the defects obey group-law fusion. We will first review the gapped boundaries of finite-group gauge theories and explain how to realize them on the lattice. Then we will present a lattice construction of the domain walls and use this construction to derive their properties, such as fusion algebra and how they transform other operators. Lastly, we will show how the results generalize to topological defects of higher codimension.

## 3.1 Codimension-one topological operators on the lattice

In this section, we will first review and define on the lattice the gapped boundaries of finite-group gauge theories, which are related to domain walls via the folding trick. Then we will present a lattice construction of the domain walls using this classification.

### 3.1.1 Gapped boundaries on the lattice

Gapped boundaries in untwisted finite-group $G$ gauge theory can be constructed from:

- Subgroup $K \leq G$.

- Topological action $\alpha \in H^{D-1}(K, U(1))$.

Given the data $(K, \alpha)$, one constructs the gapped boundary $\mathcal{B}_{K,\alpha}$ by restricting the gauge fields and gauge transformations on the boundary to be elements of $K$ and one decorates the boundary with the corresponding topological action $\alpha \in H^{D-1}(K, U(1))$ as in (2.13). The above construction is compatible and generalizes the Beigi-Shor-Whalen classification [76] of gapped boundaries in the quantum double model in $D = 3$.

### 3.1.2   Domain walls on the lattice from the folding trick

Gapped domain walls can be obtained from gapped boundaries by the folding trick. In particular, we should be able to give a constructive definition of a codimension-one domain wall of $G$ gauge theory from the data that specifies a gapped boundary of $G \times G$ gauge theory, i.e., a subgroup $H \leq G \times G$ and a topological action $\alpha \in H^{D-1}(H, U(1))$. In this section, we outline this construction.

Given a subgroup $H \leq G \times G$ and a codimension-one connected, closed and orientable submanifold $\Sigma$, we define the domain wall, $\mathcal{D}_H(\Sigma)$ by restricting the gauge group elements of the connection along $\Sigma$ to lie in the subgroup $H$ and by properly gluing the $H$ gauge group elements of $\Sigma$ with the $G$ gauge group elements of the rest of spacetime. We can further decorate the domain wall with a topological action $\alpha \in H^{D-1}(H, U(1))$ which gives the domain wall $\mathcal{D}_{H,\alpha}(\Sigma)$. In more detail, $\mathcal{D}_{H,\alpha}(\Sigma)$ is defined as:

- **Gauge field configurations:** Each edge on $\Sigma$ has group elements $(h_L, h_R) \in H \leq G \times G$ instead of $g \in G$ (where $L$ and $R$ is defined globally with respect to the orientation of the normal bundle $N\Sigma$). Because of this modification, one needs to specify the appropriate holonomy for 2-simplices that have edges both in and outside $\Sigma$, i.e., we need to define the flatness condition of (2.2) for such 2-simplices. The holonomy picks a $h_L$ (or $h_R$) contribution if the edges comes from the left (or right) of $\Sigma$. See Fig. 2 for an example of a valid flat gauge field configuration.

- **Gauge transformations:** Gauge transformation on the vertices of $\Sigma$ by $(k_L, k_R) \in H$ transforms the group elements on the edges that meet the vertex:

  - If the edge is on $\Sigma$ and pointing towards the vertex, the group element $(h_L, h_R)$ on the edge transforms into $(h_L \cdot k_L^{-1}, h_R \cdot k_R^{-1})$. If the edge is pointing away from the vertex, the group element transforms to $(k_L \cdot h_L, k_R \cdot h_R)$.
  - If the edge is outside $\Sigma$ with group element $g_L \in G$ and joins $\Sigma$ from the left, the group element transforms into $g_L \cdot k_L^{-1}$. If it leaves $\Sigma$ to the left the group element transforms into $k_L \cdot g_L$.
  - If the edge is outside $\Sigma$ with group element $g_R \in G$ and joins $\Sigma$ from the right, the group element transforms into $g_R \cdot k_R^{-1}$. If it leaves $\Sigma$ to the right the group element transforms into $k_R \cdot g_R$.

  See Fig. 3 for illustration.

- **Topological action:** The topological action $\alpha \in H^{D-1}(H, U(1))$ is evaluated for all $(D-1)$-simplices of $\Sigma$. Whenever $\alpha$ is trivial we suppress it from our notation for the domain wall. A domain wall with trivial topological action is said to be *untwisted* and *twisted* otherwise.

However, the holonomy for a contractible path that crosses $\Sigma$ is not trivial in general, for example, the path $\gamma = (v_1, v_3, v_2, v_4, v_1)$ has holonomy: $g_\gamma = g_L \cdot h_R \cdot h_L^{-1} \cdot g_L^{-1}$ which is not 1 in general. Notice that depending on the subgroup $H$, the domain wall can source holonomy for loops that pierce the wall. For example, in Fig. 2 one can check that the holonomy along the paths $(v_1, v_3, v_4)$ and $(v_2, v_3, v_4)$ are trivial. However, the holonomy for a contractible path that crosses $\Sigma$ is not trivial in general, for example, the path $\gamma = (v_1, v_3, v_2, v_4, v_1)$ has holonomy: $g_\gamma = g_L \cdot h_R \cdot h_L^{-1} \cdot g_L^{-1}$ which is not 1 in general. This feature is crucial for constructing the above domain walls as condensations where the non-trivial holonomy is generated by magnetic defect insertions [100].

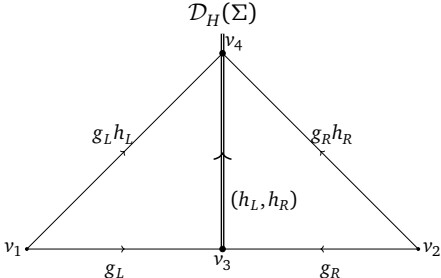

Figure 2: Example of valid flat gauge field configuration in a local region of $\mathcal{D}_H(\Sigma)$. Note that the holonomies of $(v_1, v_3, v_4, v_1)$ and $(v_2, v_3, v_4, v_2)$ are trivial, but the holonomy of $(v_1, v_3, v_2, v_4, v_1)$ is not trivial in general.

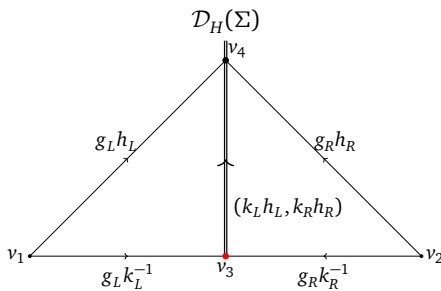

Figure 3: Example of equivalent gauge field configuration for the same local region. They are related by a gauge transformation with parameter given by $(k_L, k_R) \in H$ on $v_3$ and $1 \in H$ on $v_1, v_2, v_4$.

Because of the modification of gauge transformations along $\Sigma$, the holonomy for large loops that pierce the wall do not in general change by conjugation under gauge transformations. This means that a Wilson line inserted in such a loop is not, in general, gauge invariant. Conversely, to have a magnetic defect ending or crossing the domain wall one would need to fix the holonomy for a simplex in $\Sigma$ to be the conjugacy class of the magnetic operator, but this might not be possible depending on $H$. We are going to see that these two features can be used to define the action of untwisted domain walls in both operators.

It is straightforward to see how the above definition can be recast in the Hamiltonian formalism of the quantum double model described around equation (2.17). For instance, to define a domain wall extended along time in $\mathcal{M} = \mathbb{R}_{\text{time}} \times M_{\text{space}}$, one should change the total Hilbert space on $M_{\text{space}}$ by having a local Hilbert space $\mathbb{C}[H]$ with $H \leq G \times G$ for edges on $\Sigma_{\text{space}}$ (a codimension-one submanifold of $M_{\text{space}}$). One should then change accordingly the definition of the $A_v$ and $B_f$ terms of the Hamiltonian for vertices along $\Sigma_{\text{space}}$ and faces with edges contained within $\Sigma_{\text{space}}$.

Table 1: Above $K_L, K_R, K$ are normal subgroups of $G$ and $\phi \in \text{Aut}(G)$. In the first row, the topological action is evaluated on the right entry of $K^{(\phi)}$. More formally, as a topological action of $H^{D-1}(K^{(\phi)}, U(1))$ it is $R^*\alpha$, i.e., the pullback of $\alpha \in H^{D-1}(K, U(1))$ by $R : K^{(\phi)} \to K$ defined by $R(\phi \cdot k, k) = k$.

| Symbol | Subgroup of $G \times G$ | Local action |
|---|---|---|
| $\mathcal{D}_{K^{(\phi)}, \alpha}$ | $K^{(\phi)} \equiv \{(\phi \cdot k, k) : k \in K\}$ | $\alpha \in H^{D-1}(K, U(1))$ |
| $\mathcal{D}_{K_L \times K_R, \alpha_L \times \alpha_R}$ | $K_L \times K_R$ | $\alpha_L \times \alpha_R \in H^{D-1}(K_L \times K_R, U(1))$ |

We will focus on the domain walls corresponding to the subgroups in Table 1. Our choice for this particular subset is that they make a closed algebra under fusion. In Section 3.2.4 we are also going to work out examples of domain walls associated with the subgroup $H = G \times G$, with a non-factorized local action, i.e., a local action $\alpha \in H^{D-1}(G \times G, U(1))$ which is not of the form $\alpha_L \times \alpha_R$ with $\alpha_L, \alpha_R \in H^{D-1}(G, U(1))$. In particular, the domain wall that implements electric-magnetic duality in $\mathbb{Z}_2$ gauge theory in $D = 3$ is precisely the domain wall $\mathcal{D}_{\mathbb{Z}_2 \times \mathbb{Z}_2, \alpha}$ with the non-factorized local action $\alpha \in H^2(\mathbb{Z}_2 \times \mathbb{Z}_2, U(1)) = \mathbb{Z}_2$. More generally, in Section 3.3.3 we are going to show that defects of this form can mix electric and magnetic operators and, in this sense, generalize the electric-magnetic duality of abelian gauge theories.

### 3.1.3 Orientation-reversal of domain walls

The definition of the domain wall $\mathcal{D}_{H,\alpha}(\Sigma)$ depends on the orientation of the manifold $\Sigma$. In an orientable ambient spacetime (which we assume) an orientation of $\Sigma$ is equivalent to an orientation of the normal bundle $N\Sigma$. Orientation-reversal of $\Sigma$ flips the normal vector and exchanges the left and right of the domain wall. Thus the domain wall associated to the subgroup $H$ becomes the image of $H$ under the automorphism of $G \times G$ defined by $T(g_L, g_R) = (g_R, g_L)$. More precisely, let $\overline{\mathcal{D}}_{H,\alpha}$ be the orientation-reversal of $\mathcal{D}_{H,\alpha}$. Then $\overline{\mathcal{D}}_{H,\alpha} = \mathcal{D}_{T(H), T^*\alpha}$ where $T(H)$ denotes the image of $H \leq G \times G$ under $T$ and $T^*\alpha$ is the pullback of $\alpha \in H^{D-1}(H, U(1))$ by $T : T(H) \to H$ (here we used that $T = T^{-1}$). In particular, for the two families of subgroups of Table 1 we have:

$$\overline{\mathcal{D}}_{K^{(\phi)},\alpha} = \mathcal{D}_{(\phi(K))^{(\phi^{-1})}, \phi^{-1*}\alpha}, \qquad \overline{\mathcal{D}}_{K_L \times K_R, \alpha_L \times \alpha_R} = \mathcal{D}_{K_R \times K_L, \alpha_R \times \alpha_R}. \tag{3.1}$$

Note that reversing the orientation of $\Sigma$ (barred defect above) is the same as taking the CPT conjugate. For invertible operators, this barred operator is thus identified with the inverse and:

$$\mathcal{D} \times \overline{\mathcal{D}} = 1 \qquad \text{(invertible symmetries)}, \tag{3.2}$$

where the right-hand side denotes the identity operator. Meanwhile, for the more general non-invertible symmetries discussed here, the fusion of $\mathcal{D}$ with its CPT conjugate $\overline{\mathcal{D}}$ is not in general the identity, but rather is a condensation defect [81] and contains the identity as well as a coherent sum of other operators.[4]

## 3.2 Fusion rules of domain walls and action on gapped boundaries

In this section, we use our lattice constructions to compute the fusion of domain walls and the action of domain walls on gapped boundaries.

**Fusion of domain walls**   Given two domain walls, $\mathcal{D}_{H,\alpha}, \mathcal{D}_{H',\alpha'}$ associated with the subgroups $H, H' \leq G \times G$ and topological actions $\alpha \in H^{D-1}(H, U(1)), \alpha' \in H^{D-1}(H', U(1))$ defined on $\Sigma$, their fusion is defined by placing them "close" together and noticing that one can rewrite the insertion as a sum of other domain walls. The coefficients of the summation are partition functions of topological quantum field theories. The geometry of the two domain walls close together is that of $\Sigma \times [0, 1]$ with $\mathcal{D}_{H,\alpha}$ defined on $\Sigma \times 0$ and $\mathcal{D}_{H',\alpha'}$ on $\Sigma \times 1$. In the following computations, we are going to use a cellular decomposition of $\Sigma \times [0, 1]$ obtained from two copies of a given triangulation of $\Sigma$ by joining equivalent vertices of the two copies. See Fig. 4 for illustration.

The fusion algebra of the two classes of the domain walls presented in Table 1 is generated by the domain walls presented in Table 2.

---

[4]We note that the condensation defect can consist of operators of the same dimension as the condensation defect. In such a case, the condensation defect can act on local operators because the operators that constitute the "mesh" in the condensation defect can act on local operators. For instance, the Kramers-Wannier duality $\sigma$ in 1+1d obeys the fusion rule $\sigma \times \sigma = 1 + \psi$ in the continuum where $\bar{\sigma} = \sigma$, and $1 + \psi$ is a condensation of $\psi$, which is the $\mathbb{Z}_2$ 0-form symmetry that acts on local operators.

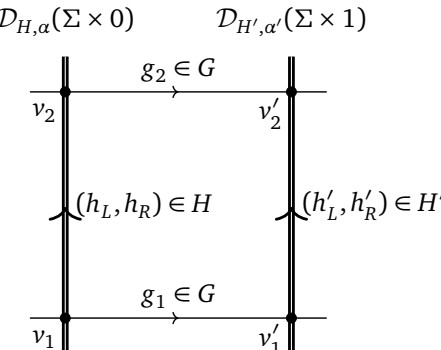

$\mathcal{D}_{H,\alpha}(\Sigma \times 0)$  $\mathcal{D}_{H',\alpha'}(\Sigma \times 1)$

Figure 4: Local region of $\Sigma$ in the presence of $\mathcal{D}_{H,\alpha}(\Sigma) \times \mathcal{D}_{H',\alpha'}(\Sigma)$. We use a cellular decomposition of $\Sigma \times [0,1]$ obtained from two copies of a given triangulation of $\Sigma$ by joining equivalent vertices of the two copies.

Table 2: Generators of the domain walls presented in Table 1. Above, $K \lhd G$ and $\phi \in \mathrm{Aut}(G)$.

| Name | Notation | Subgroup of $G \times G$ | Local action |
|---|---|---|---|
| Diagonal | $\mathcal{D}_{K^{(\mathrm{id})},\alpha}$ | $K^{(\mathrm{id})} = \{(k,k) : k \in K\}$ | $\alpha \in H^{D-1}(K, U(1))$ |
| Automorphism | $\mathcal{D}_{G^{(\phi)}}$ | $G^{(\phi)} = \{(\phi \cdot g, g) : g \in G\}$ | Trivial |
| Magnetic | $\mathcal{D}_{G \times G}$ | $G \times G$ | Trivial |

The fusion of any set of domain walls within the two classes of Table 1 can be computed using the fusion rules (derived below):

$$\mathcal{D}_{G^{(\phi)}} \times \mathcal{D}_{G^{(\phi')}} = \mathcal{D}_{G^{(\phi \circ \phi')}}, \tag{3.3}$$

$$\mathcal{D}_{G^{(\phi)}} \times \mathcal{D}_{G \times G} = \mathcal{D}_{G \times G} \times \mathcal{D}_{G^{(\phi)}} = \mathcal{D}_{G \times G}, \tag{3.4}$$

$$\mathcal{D}_{G^{(\phi)}} \times \mathcal{D}_{K^{(\mathrm{id})},\alpha} = \mathcal{D}_{K^{(\phi)},\alpha}, \tag{3.5}$$

$$\mathcal{D}_{K^{(\mathrm{id})},\alpha} \times \mathcal{D}_{G^{(\phi)}} = \mathcal{D}_{G^{(\phi)}} \times \mathcal{D}_{\phi^{-1}(K),\phi^*\alpha} = \mathcal{D}_{(\phi^{-1}(K))^{(\phi)},\phi^*\alpha}, \tag{3.6}$$

$$\mathcal{D}_{K^{(\mathrm{id})},\alpha} \times \mathcal{D}_{K'^{(\mathrm{id})},\alpha'} = \frac{|G|}{|K \cdot K'|} \mathcal{D}_{(K \cap K')^{(\mathrm{id})},\alpha \cdot \alpha'}, \tag{3.7}$$

$$\mathcal{D}_{K_L^{(\mathrm{id})},\alpha_L} \times \mathcal{D}_{G \times G} \times \mathcal{D}_{K_R^{(\mathrm{id})},\alpha_R} = \mathcal{D}_{K_L \times K_R, \alpha_L \times \alpha_R}, \tag{3.8}$$

$$\mathcal{D}_{G \times G} \times \mathcal{D}_{K^{(\mathrm{id})},\alpha} \times \mathcal{D}_{G \times G} = \mathcal{Z}(K,\alpha) \mathcal{D}_{G \times G}, \tag{3.9}$$

with $\phi \circ \phi'$ the automorphism composition of $\phi, \phi' \in \mathrm{Aut}(G)$; $\phi^{-1}(K)$ the image of $K$ under $\phi^{-1}$; $\phi^*\alpha$ the pullback of $\alpha : K^{D-1} \to U$ by $\phi : \phi^{-1}(K) \to K$; $\alpha \cdot \alpha'|_{K \cap K'} \in H^{D-1}(K \cap K', U(1))$; $|G|/|K \cdot K'|$ the 0-form partition function of $G/K \cdot K'$ gauge theory on $\Sigma$, (see the discussion around (2.9)); and $\mathcal{Z}(K,\alpha)$ the partition function of $K$ gauge theory twisted by $\alpha$ on $\Sigma$.

As an example, the second class of domain walls presented in Table 1 (the factorized domain walls) is generated by the diagonal and the magnetic domain walls. The fusion of factorized domain walls can be derived from (3.7), (3.8) and (3.9) using associativity and is:

$$\mathcal{D}_{K_L \times K_R, \alpha_L \times \alpha_R} \times \mathcal{D}_{K'_L \times K'_R, \alpha'_L \times \alpha'_R} = \frac{|G|}{|K_R \cdot K'_L|} \mathcal{Z}(K_R \cap K'_L, \alpha_R \cdot \alpha'_L) \mathcal{D}_{K_L \times K'_R, \alpha_L \times \alpha'_R}. \tag{3.10}$$

$$\mathcal{D}_{H,\alpha}(\partial\mathcal{M}\times 0) \qquad \mathcal{B}_{K,\beta}(\partial\mathcal{M}\times 1)$$

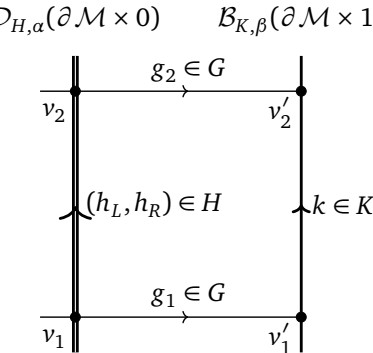

Figure 5: Local region of the boundary $\partial\mathcal{M}$ in the presence of $\mathcal{D}_{H,\alpha}(\partial\mathcal{M})\times\mathcal{B}_{K,\beta}(\partial\mathcal{M})$. We use a cellular decomposition of $\partial\mathcal{M}\times[0,1]$ obtained from two copies of a given triangulation of $\partial\mathcal{M}$ by joining equivalent vertices of the two copies.

Note that the coefficient is again a partition function on $\Sigma$: that of a $K_R\cap K'_L$ gauge theory twisted by $\alpha_R\cdot\alpha'_L$ decoupled from an untwisted $G/K_R\cdot K'_L$ zero-form gauge theory. If instead the domain walls were decorated with non-factorized topological actions $\alpha$ and $\alpha'$, the fusion coefficient would depend on the topological actions in a non-trivial way. In particular, the result would not generally be uniform in the spacetime dimension. We will give examples of this in Section 3.2.4.

**Action of domain walls on gapped boundaries** Similarly, given a domain wall $\mathcal{D}_{H,\alpha}$ and a gapped boundary $\mathcal{B}_{K,\beta}$ associated with the subgroups $H\leq G\times G$, $K\leq G$ and topological actions $\alpha\in H^{D-1}(H,U(1))$, $\beta\in H^{D-1}(K,U(1))$, one can take the domain wall to the boundary which will act on the gapped boundary generating a sum of gapped boundaries. The coefficients of the summation are partition functions of topological quantum field theories. The geometry of the domain wall action on the gapped boundary is that of $\partial\mathcal{M}\times[0,1]$ with $\mathcal{D}_{H,\alpha}$ defined along $\Sigma=\Sigma\times 0$ and $\mathcal{B}_{K,\beta}$ along $\Sigma\times 1$. Similarly to the fusion of domain walls we will use a cellular decomposition of $\partial\mathcal{M}\times[0,1]$ obtained from two copies of a given triangulation of $\partial\mathcal{M}$ by joining equivalent vertices of the two copies. See Fig. 5 for illustration.

The above definition gives the action of domain walls on gapped boundaries "from left to right". The action "from right to left" is the same as the action ("from left to right") of the orientation-reversal of the domain wall in consideration.

The domain walls of Table 2 have the following action on the gapped boundaries:

$$\mathcal{D}_{G^{(\phi)}}\times\mathcal{B}_{K,\alpha}=\mathcal{B}_{\phi(K),\phi^{-1*}\alpha}\,, \tag{3.11}$$

$$\mathcal{D}_{K^{(\mathrm{id})},\alpha}\times\mathcal{B}_{K',\alpha'}=\frac{|G|}{|K\cdot K'|}\mathcal{B}_{K\cap K',\alpha\cdot\alpha'}\,, \tag{3.12}$$

$$\mathcal{D}_{G\times G}\times\mathcal{B}_{K,\alpha}=\mathcal{Z}(K,\alpha)\mathcal{B}_{G}\,, \tag{3.13}$$

with $\phi(K)$ the image of $K\leq G$ under $\phi\in\mathrm{Aut}(G)$; $\phi^{-1*}\alpha$ the pullback of $\alpha$ by $\phi^{-1}:\phi(K)\to K$; $\alpha\cdot\alpha'|_{K\cap K'}\in H^{D-1}(K\cap K',U(1))$; and $\mathcal{Z}(K,\alpha)$ the partition function of $K$ gauge theory twisted by $\alpha$ on $\partial\mathcal{M}$.

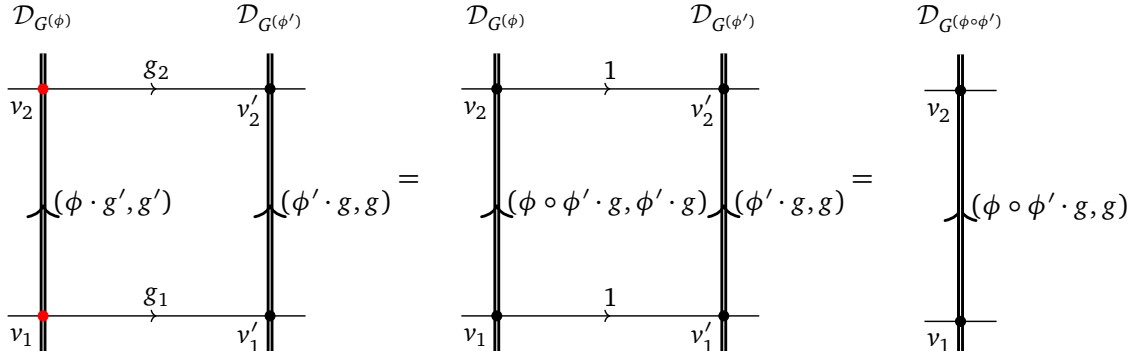

Figure 6: Derivation of the automorphism fusion rule (3.3). From the gauge transformation with image $\vec{h}(v_i) = (\phi \cdot g_i^{-1}, g_i^{-1}) \in G^{(\phi)}$ for all $v_i$ in $\mathcal{D}_{G^{(\phi)}}$ we go from a generic gauge field configuration to temporal gauge in the second figure. By the flatness condition, the group elements on the right and left of each domain wall are equal. Gauge transformations with $\vec{h}(v_i) = (\phi \cdot \phi' \cdot h, \phi' \cdot h) \in G^{(\phi)}$ and $\vec{h}(v_i') = (\phi' \cdot h, h) \in G^{(\phi')}$ preserve the temporal gauge and make the gauge transformations of $\mathcal{D}_{G^{(\phi \circ \phi')}}$.

### 3.2.1 Automorphism domain walls

In this section we derive the fusion rules (3.3) and (3.4) and the action on boundary (3.11):

$$\mathcal{D}_{G^{(\phi)}} \times \mathcal{D}_{G^{(\phi')}} = \mathcal{D}_{G^{(\phi \circ \phi')}}, \tag{3.14}$$

$$\mathcal{D}_{G^{(\phi)}} \times \mathcal{D}_{G \times G} = \mathcal{D}_{G \times G} \times \mathcal{D}_{G^{(\phi)}} = \mathcal{D}_{G \times G}, \tag{3.15}$$

$$\mathcal{D}_{G^{(\phi)}} \times \mathcal{B}_{K,\alpha} = \mathcal{B}_{\phi(K), \phi^{-1*}\alpha}. \tag{3.16}$$

involving the domain walls with automorphism subgroups $G^{(\phi)} = \{(\phi \cdot g, g) : g \in G\} \leq G \times G$ with $\phi \in \mathrm{Aut}(G)$.

We start with the first which we call the *automorphism fusion rule*. Consider the cellular decomposition of $\Sigma \times [0,1]$ illustrated in Fig. 4. From the gauge transformation with image $\vec{h}(v_i) = (\phi \cdot g_i^{-1}, g_i^{-1}) \in G^{(\phi)}$ for all $v_i$ in $\mathcal{D}_{G^{(\phi)}}$ we go from a generic gauge field configuration to one with perpendicular edges equal to the identity. We call this gauging fixing condition *temporal gauge*. By the flatness condition explained and illustrated in Fig. 2, the holonomy of the path $\gamma = (v_1, v_2, v_2', v_1', v_1)$ should be trivial, which shows that the group elements on the right and left of each domain wall are equal. Gauge transformations with $\vec{h}(v_i) = \vec{h}(v_i')$ preserve the temporal gauge and correspond to the gauge transformations of $\mathcal{D}_{G^{(\phi \phi')}}$. This shows that performing the path integral with the domain walls $\mathcal{D}_{G^{(\phi)}}$ and $\mathcal{D}_{G^{(\phi')}}$ close together is equivalent to performing the path integral with the domain wall $\mathcal{D}_{G^{(\phi \circ \phi')}}$ instead. See Fig. 6 for a summary and illustration.

The $\mathcal{D}_{G^{(\phi)}}$ defect is associated with an invertible symmetry of discrete gauge theories and fuses according to the automorphism composition. In particular:

$$\mathcal{D}_{G^{(\phi)}} \times \overline{\mathcal{D}}_{G^{(\phi)}} = \mathcal{D}_{G^{(\phi)}} \times \mathcal{D}_{G^{(\phi^{-1})}} = \mathcal{D}_{G^{(\phi \circ \phi^{-1})}} = \mathcal{D}_{G^{(\mathrm{id})}} = 1, \tag{3.17}$$

where we used (3.1) and (3.3). As we are going to see in Section 3.3, the action of the domain wall $\mathcal{D}_{G^{(\phi)}}$ on other operators is insensitive to the action of inner automorphisms (which implement global gauge transformations). If we denote by $\mathrm{Aut}(G)$ the group of all automorphisms of $G$ and by $\mathrm{Inn}(G)$ the subgroup of inner automorphisms, the physical data is the projection of $\phi \in \mathrm{Aut}(G)$ to the quotient $\mathrm{Out}(G) = \mathrm{Aut}(G)/\mathrm{Inn}(G)$ of outer automorphisms.

Therefore, generically, the ordinary symmetry of discrete gauge theories contains the subgroup $\text{Out}(G)$.

The fusion rule (3.4) and the action on gapped boundary (3.11) can be derived following the same method. The derivations are summarized in Fig. 7 and Fig. 8. The fact that one gets the pullback of $\alpha : K^{D-1} \to G$ by $\phi^{-1} : \phi(K) \to K$ follows from the fact that the topological boundary is evaluated on $K$ group elements, as shown in Fig. 8.

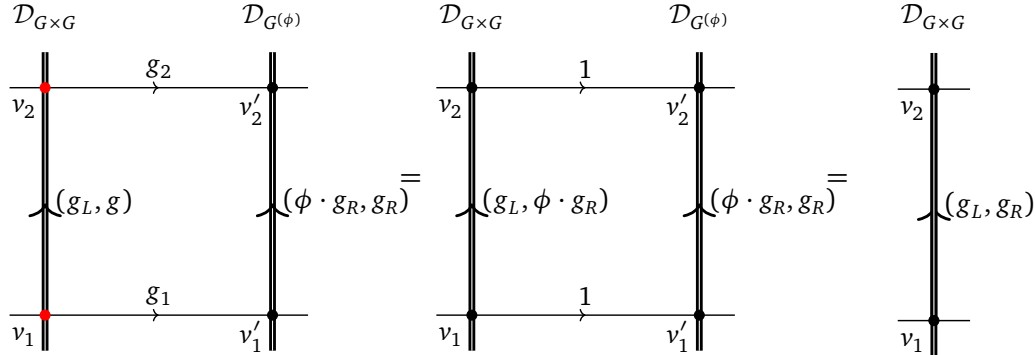

Figure 7: Derivation of the fusion rule (3.4). From the gauge transformation with image $\vec{h}(v_i) = (1, g_i^{-1}) \in G \times G$ for all $v_i$ in $\mathcal{D}_{G \times G}$ we go from a generic gauge field configuration to temporal gauge in the second figure. By the flatness condition, the group elements on the right and left of each domain wall are equal. Gauge transformations with $\vec{h}(v_i) = (h_L, \phi \cdot h_R) \in G \times G$ and $\vec{h}(v_i') = (\phi \cdot h_R, h_R) \in G^{(\phi)}$ preserve the temporal gauge and make the gauge transformations of the resulting $\mathcal{D}_{G \times G}$.

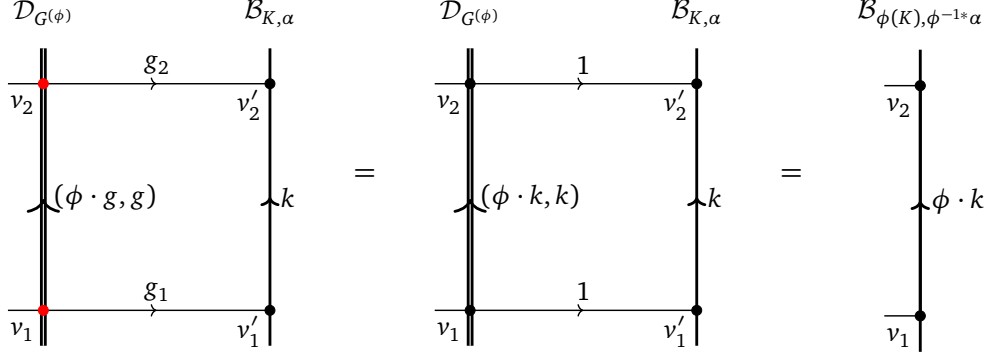

Figure 8: Derivation of the action on boundary (3.11). From the gauge transformation with image $\vec{h}(v_i) = (\phi \cdot g_i^{-1}, g_i^{-1}) \in G^{(\phi)}$ for all $v_i$ in $\mathcal{D}_{G^{(\phi)}}$ we go from a generic gauge field configuration to temporal gauge in the second figure. By the flatness condition, the group elements on the right of the domain wall and on the boundary are equal. Gauge transformations with $\vec{h}(v_i) = (\phi \cdot h, h) \in G^{(\phi)}$ and $\vec{h}(v_i') = h \in K$ preserve the temporal gauge and make the gauge transformations of the resulting boundary $\mathcal{B}_{\phi(K), \phi^{-1*}\alpha}$. Note that in the intermediate step the topological action is still evaluated on $k$ group elements, so the fusion outcome has the pullback of $\alpha$ by $\phi^{-1}$.

### 3.2.2 Diagonal domain walls

In this section we derive the fusion rules (3.5), (3.6) and (3.7) and the action on boundaries (3.12):

$$\mathcal{D}_{G^{(\phi)}} \times \mathcal{D}_{K^{(\mathrm{id})},\alpha} = \mathcal{D}_{K^{(\phi)},\alpha}, \tag{3.18}$$

$$\mathcal{D}_{K^{(\mathrm{id})},\alpha} \times \mathcal{D}_{G^{(\phi)}} = \mathcal{D}_{G^{(\phi)}} \times \mathcal{D}_{\phi^{-1}(K),\phi^*\alpha} = \mathcal{D}_{(\phi^{-1}(K))^{(\phi)},\phi^*\alpha}, \tag{3.19}$$

$$\mathcal{D}_{K^{(\mathrm{id})},\alpha} \times \mathcal{D}_{K'^{(\mathrm{id})},\alpha'} = \frac{|G|}{|K \cdot K'|} \mathcal{D}_{(K\cap K')^{(\mathrm{id})},\alpha\cdot\alpha'}, \tag{3.20}$$

$$\mathcal{D}_{K^{(\mathrm{id})},\alpha} \times \mathcal{B}_{K',\alpha'} = \frac{|G|}{|K \cdot K'|} \mathcal{B}_{K\cap K',\alpha\cdot\alpha'}, \tag{3.21}$$

associated to the automorphism subgroup defined in the previous section and the diagonal subgroups $K^{(\mathrm{id})} = \{(k,k) : k \in K\}$ with $K \lhd G$. The topological actions are elements of $\alpha \in H^{D-1}(K,U(1))$, $\alpha' \in H^{D-1}(K',U(1))$ and $\alpha \cdot \alpha'|_{K\cap K'} \in H^{D-1}(K \cap K',U(1))$.

The derivation of the first two fusion rules (3.5) and (3.6) is similar to what we did in the previous section and is summarized and illustrated in Fig. 9 and Fig. 10. We remind that we defined the domain wall $\mathcal{D}_{K^{(\phi)},\alpha}$ by having the topological action $\alpha \in H^{D-1}(K,U(1))$ evaluated on the right entry, see Table (1). This convention explains the pullback when we invert the order of multiplication.

The derivation of (3.7), which we call the *diagonal fusion rule*, has a novelty. Similarly from what we had in the automorphism derivation we find that the gauge field configuration in $\Sigma \times 0$ will give non-zero result only if the same gauge field configuration appears on $\Sigma \times 1$. This can happen just if the group elements belongs to $\in K \cap K'$. In this case, however, one needs to add the fusion coefficient $|G|/|K \cdot K'|$ that comes from summing the degrees of freedom that cannot be removed from gauge transformations. As a simple example, in the setup we are working in, we can derive the prefactor for the particular case $\mathcal{D}_1 \times \mathcal{D}_1 = |G|\mathcal{D}_1$. By using the same cellular decomposition illustrated in Fig. 4, we see that we cannot go to the temporal gauge because the gauge transformations of $\mathcal{D}_1$ are also restricted to the identity subgroup.

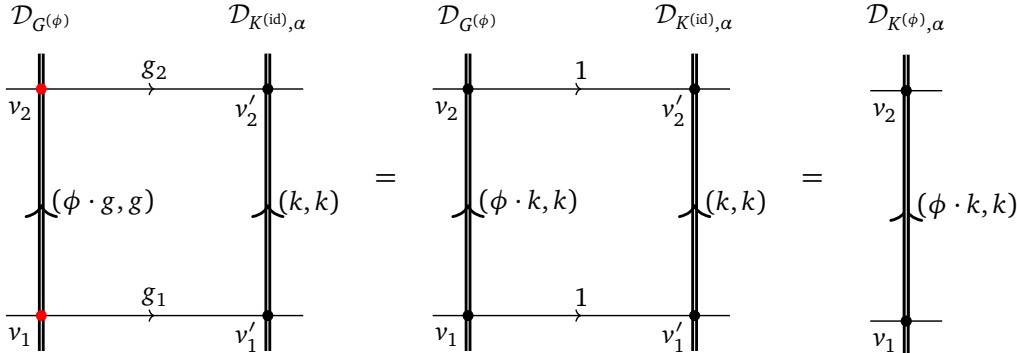

Figure 9: Derivation of the fusion rule (3.5). From the gauge transformation with parameter $\vec{h}(v_i) = (\phi \cdot g_i^{-1}, g_i^{-1}) \in G^{(\phi)}$ for all $v_i$ in $\mathcal{D}_{G^{(\phi)}}$ we go from a generic gauge field configuration to temporal gauge in the second figure. By the flatness condition, the group elements on the right and left of each domain wall are equal. Gauge transformations with $\vec{h}(v_i) = (\phi \cdot h, h) \in G^{(\phi)}$ and $\vec{h}(v_i') = (h, h) \in K^{(\mathrm{id})}$ preserve the temporal gauge and make the gauge transformations of the resulting $\mathcal{D}_{K^{(\phi)},\alpha}$. The topological action is evaluated on $K$ group elements in all steps so we do not have the pullback of $\alpha$ by $\phi$.

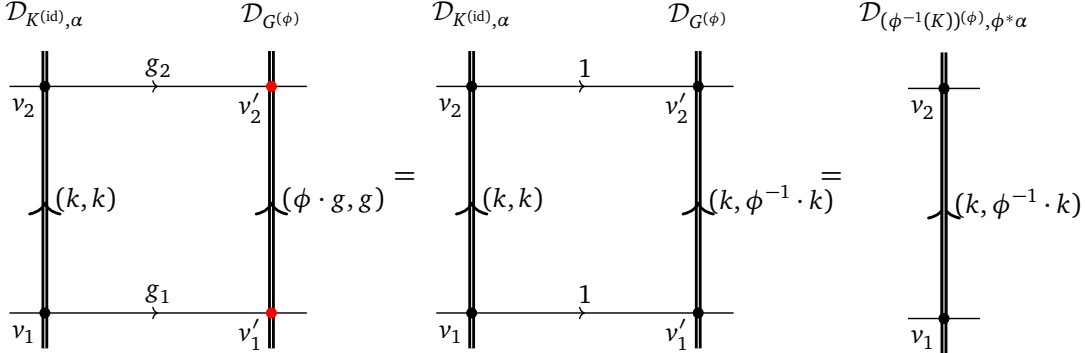

Figure 10: Derivation of the fusion rule (3.6). From the gauge transformation with parameter $\vec{h}(v_i') = (g_i, \phi^{-1} \cdot g_i) \in G^{(\phi)}$ for all $v_i$ in $\mathcal{D}_{G^{(\phi)}}$ we go from a generic gauge field configuration to temporal gauge in the second figure. By the flatness condition, the group elements on the right and left of each domain wall are equal. Gauge transformations with $\vec{h}(v_i') = (h, \phi^{-1} \cdot h) \in G^{(\phi)}$ and $\vec{h}(v_i) = (h, h) \in K^{(\text{id})}$ preserve the temporal gauge and make the gauge transformations of the resulting $\mathcal{D}_{(\phi^{-1}(K))^{(\phi)}, \phi^*\alpha}$. The topological action is evaluated on $K$ group elements in the intermediate step so we need the pullback of $\alpha$ by $\phi : \phi^{-1}(K) \to K$.

But by the flatness condition, the group elements on the perpendicular edges should be equal. By summing over all these configurations we get the factor of $|G|$. See Fig. 11 for illustration.

The general case can be argued similarly, the difference is that any element $g$ in the middle that can be written in the form $kk'$ with $k \in K$ and $k' \in K'$ can be trivialized by gauge transformations that are not part of the $K \cap K'$ gauge transformations. This means, that the pre-factor is $|G|/|K \cdot K'|$ instead of $|G|$. Note that the fusion rule is consistent with the fact that $\mathcal{D}_{G^{(\text{id})}} = 1$. Note that the prefactor is an integer because for normal subgroups $K \cdot K'$ is also a subgroup and by Lagrange's theorem it divides the order of $G$.

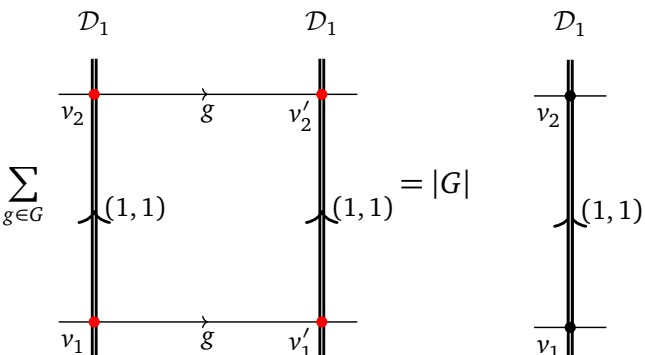

Figure 11: Derivation of the fusion rule $\mathcal{D}_1 \times \mathcal{D}_1 = |G|\mathcal{D}_1$. The red dots highlights the source of gauge transformations that are elements of $1 \leq G \times G$ and cannot change the holonomy along the perpendicular edge. By the flatness condition, the group elements on the perpendicular edges are equal. Performing the summation of the middle horizontal edge gives rise to the $|G|$ factor.

If the two domain walls $\mathcal{D}_{K^{(\text{id})}}, \mathcal{D}_{K'^{(\text{id})}}$ are decorated with additional topological term $\alpha \in H^{D-1}(K, U(1)), \alpha' \in H^{D-1}(K', U(1))$, their fusion result has decoration given by attaching $\alpha \cdot \alpha'|_{K \cap K'} \in H^{D-1}(K \cap K', U(1))$:

$$\mathcal{D}_{K^{(\text{id})},\alpha}(\Sigma) \times \mathcal{D}_{K'^{(\text{id})},\alpha'}(\Sigma) = \frac{|G|}{|K \cdot K'|} \mathcal{D}_{(K \cap K')^{(\text{id})},\alpha \cdot \alpha'}(\Sigma). \tag{3.22}$$

This is more easily seen by thinking of $\mathcal{D}_{K^{(\text{id})},\alpha}(\Sigma)$ as the product of $\mathcal{D}_{K^{(\text{id})}}(\Sigma)$ and the invertible electric defect $W_\alpha(\Sigma)$ defined in (2.13). The action of $\mathcal{D}_{K^{(\text{id})},\alpha}$ on gapped boundaries (3.12) can be derived in a similar way.

Let us make a few remarks:

- By the lattice definition $\mathcal{D}_{G^{(\text{id})}} = 1$ is the identity defect and $\mathcal{D}_{G^{(\text{id})},\alpha}(\Sigma)$ is obtained by decorating $\Sigma$ with the topological action $\alpha \in H^{D-1}(G, U(1))$, which is the codimension-one invertible electric defect defined in (2.13). One can see that the fusion of these defects is the group multiplication in $H^{D-1}(G, U(1))$. This is consistent with the property that fusing such domain walls is the same as first stacking the SPT phases with $G$ symmetry labelled by $\alpha, \alpha'$ on the wall and then gauging the $G$ symmetry [28]. Generically, the full ordinary symmetry of discrete gauge theory is generated by $\mathcal{D}_{G^{(\phi)}}$ and $\mathcal{D}_{G^{(\text{id})},\alpha}$ which form the group:

$$\text{Out}(G) \ltimes H^{D-1}(G, U(1)). \tag{3.23}$$

In $D = 3$, Wilson lines and magnetic defects are both lines, so there exist other ordinary symmetries that consistently permute the Wilson lines and magnetic defects. Electric-magnetic duality symmetry in $\mathbb{Z}_2$ is such an example. It corresponds to the domain wall associated with the subgroup $H = \mathbb{Z}_2 \times \mathbb{Z}_2 \leq \mathbb{Z}_2 \times \mathbb{Z}_2$ and the non-factorized topological action $\alpha_2 \in H^2(\mathbb{Z}_2 \times \mathbb{Z}_2, U(1)) = \mathbb{Z}_2$.

- The fusion of diagonal domain walls is commutative. In Section 3.4, we will generalize them to higher-codimension topological defects where the fusion rules must be commutative [82]. This is to be contrasted with the factorized and automorphism domain walls, which obey a non-commutative algebra, and therefore cannot be generalized to higher-codimension.

- As we are going to see in Section 4, the fusion coefficient can be interpreted as the partition function of Stueckelberg scalars, which is equal to the 0-form partition function of untwisted $G/K \cdot K'$ gauge theory, (2.9). In particular, the coefficient is sensitive to the topology of $\Sigma$, more precisely, to $\pi_0(\Sigma)$, which here we have assumed to be 1.

### 3.2.3 Factorized domain walls

In this section we first derive the elementary fusion rules (3.8) and (3.9), and the action on boundaries (3.13):

$$\mathcal{D}_{K_L^{(\text{id})},\alpha_L} \times \mathcal{D}_{G \times G} \times \mathcal{D}_{K_R^{(\text{id})},\alpha_R} = \mathcal{D}_{K_L \times K_R, \alpha_L \times \alpha_R}, \tag{3.24}$$

$$\mathcal{D}_{G \times G} \times \mathcal{D}_{K_R^{(\text{id})},\alpha} \times \mathcal{D}_{G \times G} = \mathcal{Z}(K, \alpha) \mathcal{D}_{G \times G}, \tag{3.25}$$

$$\mathcal{D}_{G \times G} \times \mathcal{B}_{K,\alpha} = \mathcal{Z}(K, \alpha) \mathcal{B}_G, \tag{3.26}$$

with $\mathcal{Z}(K, \alpha)$ the partition function of $K$ gauge theory twisted by $\alpha \in H^{D-1}(K, U(1))$ on $\Sigma$. Then, we use these results to derive (3.10).

The factorized domain walls are generated by the diagonal domain walls and $\mathcal{D}_{G \times G}$ from the first equation, (3.8). To derive this fusion we proceed similarly to what we did before by

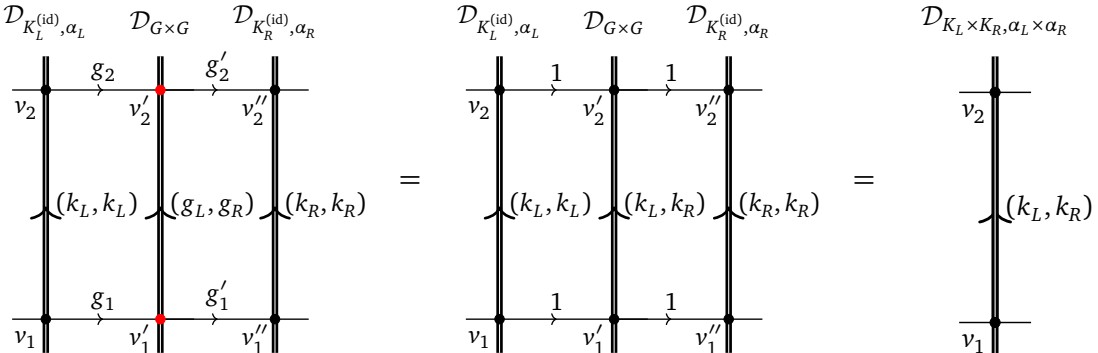

Figure 12: Derivation of the factorized fusion rule (3.8). The first two figures are related by a gauge transformation with $\vec{h}(v_i') = (g_i, g_i'^{-1}) \in G \times G$ for all $v_i'$ on $\mathcal{D}_{G \times G}$. By the flatness condition the vertical edge on the middle has group elements $(k_L, k_R)$. Gauge transformations with $\vec{h}(v_i) = (h_L, h_L) \in K_L^{(id)}$, $\vec{h}(v_i') = (h_L, h_R) \in G \times G$ and $\vec{h}(v_i'') = (h_R, h_R) \in K_R^{(id)}$ preserve the temporal gauge and make the gauge transformations of $\mathcal{D}_{K_L \times K_R}$. The topological actions are carried along the way.

using the cellular decomposition of Fig. 4. First, we perform gauge transformations on the vertices of $\mathcal{D}_{G \times G}$ to get to temporal gauge; then we use the flatness condition to see that the elements on the middle layer are $(k_L, k_R) \in K_L \times K_R$; and finally we note that the remaining gauge transformations are elements of $K_L \times K_R$. The topological actions are carried along the way. See Fig. 12 for an illustration.

Now, we derive the second fusion rule, (3.9). The result follows from noticing that the $\mathcal{D}_{G \times G}$ domain wall cuts the manifold into two disconnected and independent components. Therefore, one can perform the partition summation for the disconnected part in the middle. This gives the $\mathcal{Z}(K, \Sigma, \alpha)$ partition function because the middle layer is topologically $\Sigma \times [0, 1]$ and retracts to $\Sigma$ with gauge group elements in $K$ and a topological action $\alpha \in H^{D-1}(K, U(1))$. Conversely, we could follow the same steps as before which are summarized in Fig. 13.

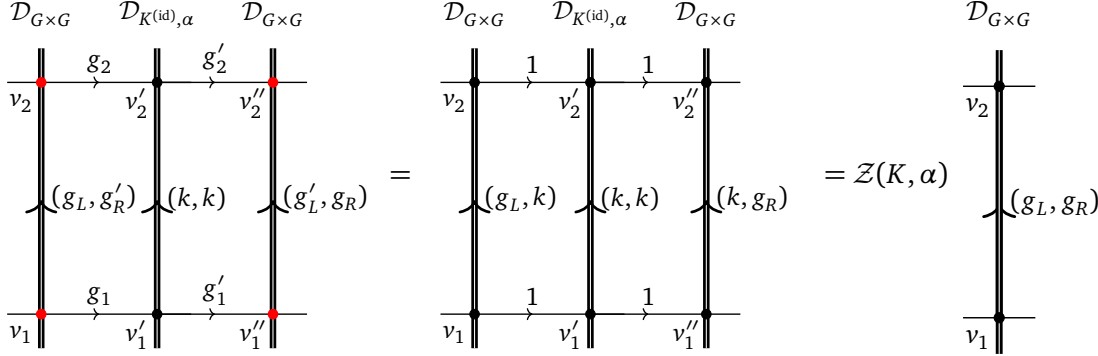

Figure 13: Derivation of the factorized fusion rule (3.9). The first two figures are related by a gauge transformation with $\vec{h}(v_i) = (1, g_i^{-1}) \in G \times G$ and $\vec{h}(v_i'') = (g_2', 1) \in G \times G$ for all vertices $v_i$ and $v_i''$ in the two $\mathcal{D}_{G \times G}$. Gauge transformations with $\vec{h}(v_i) = (1, h) \in G \times G$, $\vec{h}(v_i') = (h, h) \in K^{(id)}$ and $\vec{h}(v_i'') = (h, 1) \in G \times G$ make the gauge transformations of the decoupled partition function obtained by summing over $k$. Gauge transformations with $\vec{h}(v_i) = (h_L, 1) \in G \times G$, $\vec{h}(v_i'') = (1, h_R) \in G \times G$ preserve the temporal gauge and make the gauge transformations of the resulting $\mathcal{D}_{G \times G}$.

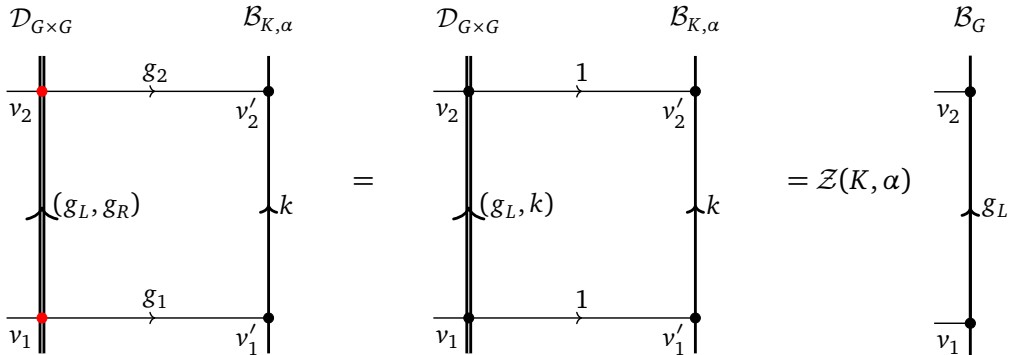

Figure 14: Derivation of the action on boundary (3.11). From the gauge transformation with image $\vec{h}(v_i) = (1, g_i^{-1}) \in G \times G$ for all $v_i$ in $\mathcal{D}_{G \times G}$ we go from a generic gauge field configuration to temporal gauge in the second figure. By the flatness condition the group elements on the right of the domain wall and on the boundary are equal. Gauge transformations with $\vec{h}(v_i) = (1, h) \in G \times G$ and $\vec{h}(v_i') = h \in K$ preserve the temporal gauge and make the gauge transformations of the decoupled partition function $\mathcal{Z}(K, \alpha)$. Gauge transformations with $\vec{h}(v_i) = (h, 1) \in G \times G$ make the gauge transformations of the resulting boundary $\mathcal{B}_G$.

The action of $\mathcal{D}_{G \times G}$ on gapped boundaries (3.13) can be derived following the same method. The derivation is summarized in Fig. 14.

The fusion of factorized domain walls (3.10) can be obtained from (3.7), (3.8) and (3.9) using associativity. For the case with trivial topological action, we have:

$$
\begin{aligned}
\mathcal{D}_{K_L \times K_R} \times \mathcal{D}_{K_L' \times K_R'} &= \mathcal{D}_{K_L^{(\mathrm{id})}} \times \mathcal{D}_{G \times G} \times \mathcal{D}_{K_R^{(\mathrm{id})}} \times \mathcal{D}_{K_L'^{(\mathrm{id})}} \times \mathcal{D}_{G \times G} \times \mathcal{D}_{K_R'^{(\mathrm{id})}} \\
&= \frac{|G|}{|K_L \cdot K_R|} \mathcal{D}_{K_L^{(\mathrm{id})}} \times \mathcal{D}_{G \times G} \times \mathcal{D}_{(K_R \cap K_L')^{(\mathrm{id})}} \times \mathcal{D}_{G \times G} \times \mathcal{D}_{K_R'^{(\mathrm{id})}} \\
&= \frac{|G|}{|K_R \cdot K_L'|} \mathcal{Z}(K_R \cap K_L') \mathcal{D}_{K_L^{(\mathrm{id})}} \times \mathcal{D}_{G \times G} \times \mathcal{D}_{K_R'^{(\mathrm{id})}} \\
&= \frac{|G|}{|K_R \cdot K_L'|} \mathcal{Z}(K_R \cap K_L') \mathcal{D}_{K_L \times K_R'}.
\end{aligned}
\tag{3.27}
$$

The derivation for the case with factorized topological action is analogous. We remark that the trivial subgroup domain wall is an example of both factorized and diagonal domain walls. One can easily check that (3.10) is equal to (3.7) when $K_R = K_L = K_R' = K_L' = 1$.

### 3.2.4 Twisted domain wall: Electric-magnetic duality

To close this section we will show how to compute the fusion and action on gapped boundaries of twisted domain walls $\mathcal{D}_{G \times G, \alpha}$ with $\alpha \in H^{D-1}(G \times G, U(1))$ a non-trivial and non-factorized local action. The result depends on the dimension and gauge group.

The derivation of the fusion $\mathcal{D}_{G \times G, \alpha} \times \mathcal{D}_{G \times G, \alpha'}$ and the action $\mathcal{D}_{G \times G, \alpha} \times \mathcal{B}_{K, \beta}$ is very similar to the derivation of $\mathcal{D}_{G \times G} \times \mathcal{D}_{G \times G} = \mathcal{Z}(G) \mathcal{D}_{G \times G}$ and $\mathcal{D}_{G \times G} \times \mathcal{B}_{K, \beta} = \mathcal{Z}(K, \beta) \mathcal{B}_G$ summarized in Fig. 13 and Fig. 14 respectively. We start with the cellular decomposition of Fig. 4 and Fig. 5, we perform a gauge transformation to get to temporal gauge and by the flatness condition, the elements on the right and left are equal. Now, however, the summation over the intermediate gauge group elements does not give a partition function because the topological action is partially evaluated on them. The answer depends on the explicit expression for the cocycles which depends on the dimension and the gauge group.

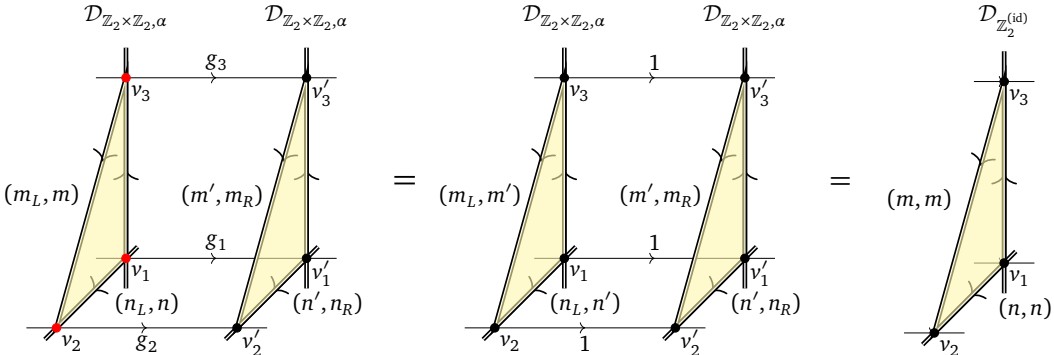

Figure 15: Derivation of the fusion rule (3.28). The first two figures are related by a gauge transformation with $\vec{h}(v_i) = (1, g_i^{-1}) \in \mathbb{Z}_2 \times \mathbb{Z}_2$ for all vertices $v_i$ in $\Sigma \times 0$. By the flatness condition the holonomy of $(v_1, v_2, v_2', v_1', v_1)$ and $(v_2, v_3, v_3', v_2', v_2)$ should be trivial, which shows that the outer vertical edges have group elements $(m_L, m')$, $(m', m_R)$, $(n_L, n')$ and $(n', n_R)$. Summing over $n'$ and $m'$ with the two topological actions forces $m_L = m_R = m$ and $n_L = n_R = n$, see (3.32), which corresponds to the $\mathcal{D}_{\mathbb{Z}_2^{(\mathrm{id})}} = 1$ domain wall.

Let us carry this procedure in detail for $G = \mathbb{Z}_2$ in $D = 3$. Because $H^2(\mathbb{Z}_2, U(1)) = 1$, this theory has just two gapped boundaries $\mathcal{B}_{\mathbb{Z}_2}$ (Neumann) and $\mathcal{B}_1$ (Dirichlet). However, $H^2(\mathbb{Z}_2 \times \mathbb{Z}_2, U(1)) = \mathbb{Z}_2$ so we can consider a domain wall twisted by a non-factorized action. We are going to show that

$$\mathcal{D}_{\mathbb{Z}_2 \times \mathbb{Z}_2, \alpha_2} \times \mathcal{D}_{\mathbb{Z}_2 \times \mathbb{Z}_2, \alpha_2} = 1, \tag{3.28}$$

$$\mathcal{D}_{\mathbb{Z}_2 \times \mathbb{Z}_2, \alpha_2} \times \mathcal{B}_{\mathbb{Z}_2} = \mathcal{B}_1, \tag{3.29}$$

$$\mathcal{D}_{\mathbb{Z}_2 \times \mathbb{Z}_2, \alpha_2} \times \mathcal{B}_1 = \mathcal{B}_{\mathbb{Z}_2}, \tag{3.30}$$

where $\alpha_2$ the non-trivial 2-cocycle of $H^2(\mathbb{Z}_2 \times \mathbb{Z}_2, U(1)) = \mathbb{Z}_2$.

Let us start with the derivation of the fusion (3.28), illustrated in Fig. 15. We need to show that the summation over the middle group elements (that would lead to $\mathcal{Z}(\mathbb{Z}_2)$ in the trivial $\alpha$ case) gives a delta function $\delta(m_L - m_R)\delta(n_L - n_R)$. To show that we need the explicit form for the algebraic cocycle $\alpha \in H^2(\mathbb{Z}_2 \times \mathbb{Z}_2, U(1))$ which has representative:

$$\alpha(n_L, n_R, m_L, m_R) = e^{\pi i n_L m_R}. \tag{3.31}$$

For simplicity, let the surface in consideration be a torus. The product of local actions over 2-simplices for a torus with $\alpha \in H^2(G, U(1))$ is equal to $\frac{\alpha(g,h)}{\alpha(h,g)}$ for $g, h \in G$. Therefore, in the configuration of Fig. 15 the total action is

$$\frac{\alpha(n_L, n', m_L, m')\alpha(n', n_R, m', m_R)}{\alpha(m_L, m', n_L, n')\alpha(m', m_R, n', n_R)} = e^{\pi i (n_L m' - m_L n' + n' m_R - m' n_R)}, \tag{3.32}$$

where we used (3.31). If we sum over $n'$ and $m'$, then we will get $\delta(m_L - m_R)\delta(n_L - n_R)$ that forces $n_L = n_R = n$ and $m_L = m_R = m$. We see that the fusion results in $\mathcal{D}_{\mathbb{Z}_2^{(\mathrm{id})}} = 1$.

The derivation of the action on the gapped boundaries is very similar and is illustrated in Fig. 16 for the $\mathcal{B}_{\mathbb{Z}_2}$ case. We need to show that the summation over the middle group elements gives the delta function $\delta(m_L)\delta(n_L)$. Using again the total action for the torus we have in this case:

$$\frac{\alpha(n_L, n', m_L, m')}{\alpha(m_L, m', n_L, n')} = e^{\pi i (n_L m' - m_L n')}, \tag{3.33}$$

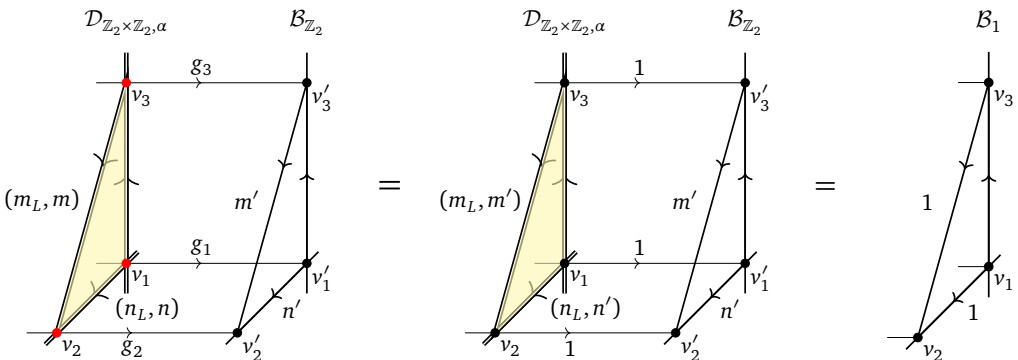

Figure 16: Derivation of the action on boundary (3.29). The first two figures are related by a gauge transformation with $\vec{h}(v_i) = (1, g_i^{-1}) \in \mathbb{Z}_2 \times \mathbb{Z}_2$ for all vertices $v_i$ in $\Sigma \times 0$. By the flatness condition the holonomy of $(v_1, v_2, v_2', v_1', v_1)$ and $(v_2, v_3, v_3', v_2', v_2)$ should be trivial, which shows that the outer vertical edges have group elements $(m_L, m)$, $m$, $(n_L, n)$ and $n$. Summing over $n'$ and $m'$ with the topological actions forces $m_L = n_L = 1$, see (3.33), which corresponds to the $\mathcal{B}_1$ gapped boundary.

which gives $\delta(m_L)\delta(n_L)$ by summing over $n'$ and $m'$. We see the action results in $\mathcal{B}_1$. The action of the domain wall on $\mathcal{B}_1$ can be derived in the same way. In this case however, the gauge group elements on the boundary are trivial which trivializes the topological action (3.31) so the result is the same as the action $\mathcal{D}_{\mathbb{Z}_2 \times \mathbb{Z}_2} \times \mathcal{B}_1 = \mathcal{B}_{\mathbb{Z}_2}$. Consistently, we could compute the action on $\mathcal{B}_1$ using the fusion (3.28), the action (3.29) and associativity which gives:

$$\mathcal{D}_{\mathbb{Z}_2 \times \mathbb{Z}_2, \alpha_2} \times \mathcal{B}_1 = \mathcal{D}_{\mathbb{Z}_2 \times \mathbb{Z}_2, \alpha_2} \times \mathcal{D}_{\mathbb{Z}_2 \times \mathbb{Z}_2, \alpha_2} \times \mathcal{B}_{\mathbb{Z}_2} = \mathcal{B}_{\mathbb{Z}_2} \,. \tag{3.34}$$

The only invertible symmetry of $\mathbb{Z}_2$ gauge theory is electric-magnetic duality, therefore $\mathcal{D}_{\mathbb{Z}_2 \times \mathbb{Z}_2, \alpha}$ is the electric-magnetic duality symmetry defect. As we are going to show in Section 3.3.3, operators with the form $\mathcal{D}_{G \times G, \alpha}$ with $\alpha$ non-factorized, generically mix invertible electric operators (2.13) and magnetic defects. Note that the same procedure could be carried out in higher dimensions and for generic gauge groups.

## 3.3 Transformation of Wilson lines and magnetic defects

When we move a topological defect across another operator, the operator can be transformed into another one. In this section, we study the transformation of other operators when they cross the gapped domain walls.

In the following, we will investigate the transformation of Wilson lines and magnetic defects. We start discussing which operators can end on untwisted gapped boundaries $\mathcal{B}_K$ of $G$ gauge theory. Then, we investigate the action of domain walls on other operators separating the discussion into: untwisted domain walls, $\mathcal{D}_H$, and twisted domain walls $\mathcal{D}_{H,\alpha}$ with $\alpha \in H^{D-1}(H, U(1))$ a non-factorized action. In the first class of domain walls, the electric and magnetic operators transform separately, while the second class of domain walls can mix them, and in this sense generalizes electric-magnetic duality. Similar distinctions are discussed in [10].

In our setup, the transformation is derived from the fact that pairs of Wilson lines and magnetic defects can be made gauge invariant if they end on the same locus of the domain wall. We exhibit the collection of all such allowed junctions via the action of the domain wall on simple Wilson lines and magnetic defects:

$$\mathcal{D}_{K,\alpha} \cdot W_{\rho_R} = \sum_{i_L \in \text{irrep}} c_{i_L} W_{\rho_{i_L}} \,, \qquad \mathcal{D}_{K,\alpha} \cdot M_{g_R} = \sum_{[g_L] \in \text{Cl}(G)} c_{g_L} M_{g_L} \,, \tag{3.35}$$

with $c_{i_L}$ and $c_{g_L}$ larger than zero if there exists a gauge invariant configuration.

### 3.3.1 Untwisted gapped boundaries

As shown in Section 3.1.1, gapped boundaries in untwisted $G$ gauge theory can be constructed by $(K, \alpha)$ with $K \leq G$ and topological action $\alpha \in H^{D-1}(K, U(1))$. Here we restrict to the case with trivial topological action. The gapped boundary $\mathcal{B}_K$ associated with the subgroup $K \leq G$ is obtained by restricting the gauge group elements on the boundary to be elements of $K$. From this definition, it follows that:

- **Wilson lines** the Wilson line $W_\rho$ can end on the gapped boundary $\mathcal{B}_K$ if the decomposition of $\rho$ in representations of $K$ contains the singlet. The number of possible junctions is equal to the multiplicity of the trivial representation in the restriction.

- **Magnetic defects** the magnetic defect $M_g$ can end on the domain wall if there exists a group element $k \in [g]$ such that $k \in K$. The number of possible junctions is equal to the number of different conjugacy classes of $K$ of such group elements.

  For illustration, consider $G = S_3$ (the symmetric group on 3 elements), the conjugacy class $[(123)]_{S_3} = \{(123), (132)\}$, and the gapped boundary $\mathcal{B}_{A_3}$ with $A_3 = \{1, (123), (132)\} \leq S_3$ (the alternating group on 3 elements). The magnetic defect $M_{(123)}$ can end on $\mathcal{B}_{A_3}$ because $(123), (132) \in A_3$ and the multiplicity in this case is 2 because $[(123)]_{A_3} = \{(123)\}$ and $[(132)]_{A_3} = \{(132)\}$ are different conjugacy classes of $A_3$.

In particular, all Wilson lines can end on $\mathcal{B}_1$ with multiplicity equal to the dimension of the corresponding irreducible representation, and all magnetic defects can end on $\mathcal{B}_G$ with multiplicity equal to one.

### 3.3.2 Untwisted domain walls

Let us begin with the untwisted domain walls $\mathcal{D}_H$. For such domain walls, the Wilson lines and the magnetic defects transform separately:

- **Wilson lines** the Wilson line $W_{\rho_L}$ can be transformed into $W_{\rho_R}$ if the decomposition of $\rho_L \otimes \overline{\rho}_R$ into representations of $H$ contains the trivial representation. If the decomposition of $\rho_L \otimes \overline{\rho}_R$ into representations of $H$ contains the trivial representation, then a configuration with $W_{\rho_L}$ and $W_{\rho_R}$ joining on $\mathcal{D}_H$ can be made gauge invariant. Again, the coefficient of the trivial representation gives the number of possible junctions.

- **Magnetic defects** the magnetic defect $M_{g_L}$ can be transformed into $M_{g_R}$ if there exists a group element $(k_L, k_R) \in [g_L] \times [g_R]$ such that $(k_L, k_R) \in H$. The number of possible junctions equals the number of different conjugacy classes of $H$ that such group elements form.

Let us give some examples to illustrate these transformations.

**Example: automorphism domain wall**  For instance, consider the domain wall that generates automorphism $\phi$, it corresponds to the subgroup $G^{(\phi)} = \{(\phi \cdot g, g) : g \in G\} \leq G \times G$. Let us consider the transformation of the magnetic defect $M_{g_L}$ into $M_{g_R}$ and the transformation of Wilson lines $W_{\rho_L}$ into $W_{\rho_R}$.

- The pair of magnetic defects $M_{g_L}, M_{g_R}$ can have a junction on the wall if and only if $\phi(g_L) = g_R$, where we extend the action of automorphism on the conjugacy class. In other words, the magnetic defect associated with the conjugacy class of $g$ transforms into $\phi^{-1}(g)$ after it passes through the domain wall from left to right. Conversely, it transforms into $\phi(g)$ after it passes through the domain wall from right to left. The multiplicity in this case is one.

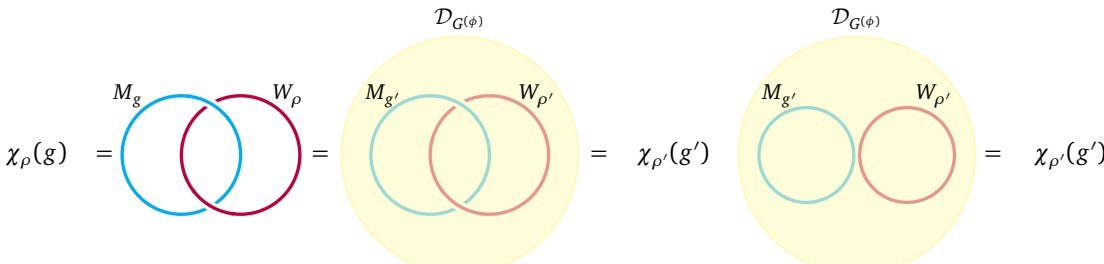

$$\chi_\rho(g) \quad = \quad \text{(figure)} \quad = \quad \text{(figure)} \quad = \quad \chi_{\rho'}(g') \quad = \quad \text{(figure)} \quad = \quad \chi_{\rho'}(g')$$

Figure 17: A linked configuration of Wilson line, $W_\rho$ and magnetic defect $M_g$, can be unlinked and contracted giving the associated character $\chi_\rho(g)$. If one nucleates an ordinary symmetry defect $\mathcal{D}_{G^{(\phi)}}$ (oriented outwards) and embed the linked pair inside it, they will be permuted to $W_{\rho'}$ and $M_{g'}$ (where we assumed that the symmetry is invertible). Unlinking the permuted pair inside the symmetry defect and contracting all operators gives the permuted character $\chi_{\rho'}(g')$. These topological manipulations shows that if $W_\rho \mapsto W_{\rho \cdot \phi^{-1}}$, then $M_g \mapsto M_{\phi(g)}$.

- The decomposition of the representation $\rho_L \otimes \overline{\rho}_R$ of $G \times G$ into representations of the subgroup $G^{(\phi)}$ contains the trivial representation when $\rho_L = \rho_R \circ \phi^{-1}$, where $\rho_R \circ \phi^{-1}$ is the representation satisfies $\rho_R \circ \phi^{-1}(g) = \rho_R(\phi^{-1} \cdot g)$ for $g \in G$. In other words, the Wilson line associated with the representation $\rho$ transforms into $\rho \circ \phi$ after it passes through the domain wall from left to right. Conversely, it transforms into $\rho \circ \phi^{-1}$ after it passes through the domain wall from right to left.

Using the notation defined in (3.35) the above can be summarized as

$$\mathcal{D}_{G^{(\phi)}} \cdot W_{\rho_i} = W_{\rho_i \circ \phi^{-1}}, \tag{3.36}$$

$$\mathcal{D}_{G^{(\phi)}} \cdot M_g = M_{\phi(g)}, \tag{3.37}$$

for every irreducible representation $\rho_i$ and conjugacy class $[g]$. These transformation properties are consistent with the fusion of automorphism domain walls (3.3). We remark that the transformation preserves the Aharonov-Bohm braiding between the Wilson lines and the magnetic defect, which is given by evaluating the character of the representation of the Wilson line on the conjugacy class of the magnetic defect. See Fig. 17 for an illustration.

**Example: diagonal domain wall with $H = 1$**   Let us consider the domain wall with $H = 1$. In this case, every Wilson line can end on the domain wall. In particular, the trivial Wilson line transforms into the direct sum of all other Wilson lines. An example is $G = \mathbb{Z}_2$ in $D = 3$, where the domain wall is called the Cheshire string domain wall [64]. No configuration of nontrivial magnetic defects can end on the domain wall with $H = 1$. This happens because it is impossible to fix the holonomy on the simplices of $\Sigma$ to be an element that is not the identity. Using the notation defined in (3.35) the above can be summarized as

$$\mathcal{D}_1 \cdot W_{\rho_i} = \sum_{k \in \text{irreps}} d_i d_k W_{\rho_k}, \tag{3.38}$$

$$\mathcal{D}_1 \cdot M_g = 0, \tag{3.39}$$

for every irreducible representation $\rho_i$ and conjugacy class $[g]$. Above, $d_i$ is the dimension of the irreducible representation $\rho_i$. These transformation properties are consistent with the fusion of diagonal domain walls (3.7).

**Example: factorized domain wall with $H = G \times G$**   This case is complementary to the previous example. No Wilson line can end and all magnetic defects can. Using the notation defined in (3.35) the above can be summarized as

$$\mathcal{D}_{G \times G} \cdot W_{\rho_i} = 0 \,, \tag{3.40}$$

$$\mathcal{D}_{G \times G} \cdot M_g = \sum_{[k] \in \mathrm{Cl}(G)} M_k \,, \tag{3.41}$$

for every irreducible representations $\rho_i$ and conjugacy class $[g]$. These transformation properties are consistent with the fusion of factorized domain walls (3.9).

### 3.3.3   Twisted domain wall: Electric-magnetic duality

In this section we show how to understand the action of $\mathcal{D}_{G \times G, \alpha}$ on other operators with $\alpha \in H^{D-1}(G \times G, U(1))$ a non-factorized topological action, i.e., $\alpha \neq \alpha_L \times \alpha_R$ with $\alpha_L, \alpha_R \in H^{D-1}(G, U(1))$. The underlying mechanism determining the action is reminiscent of anomaly inflow [83].

As shown in (3.41), all magnetic defects can terminate on $\mathcal{D}_{G \times G}$. However, if the domain wall is decorated with a non-factorized topological action $\alpha \in H^{D-1}(G \times G, U(1))$, such junctions are no longer gauge invariant. Conversely, an invertible electric operator (2.13) cannot end on $\mathcal{D}_{G \times G}$ because it is not invariant under gauge transformations. Interestingly, the two effects can cancel and a configuration with an invertible electric operator ending on the same locus as a magnetic defect can be made gauge invariant.

Let us illustrate this generic feature using again the simplest example of $G = \mathbb{Z}_2$ gauge theory in $D = 3$. With no other insertions, the Wilson line $W$ coming from the left of $\mathcal{D}_{\mathbb{Z}_2 \times \mathbb{Z}_2, \alpha_2}$ cannot end on a vertex $v_0$ of the domain wall because under a gauge transformation with parameter $\vec{h}(v_0) = (1,0) \in \mathbb{Z}_2 \times \mathbb{Z}_2$ the Wilson line would change to

$$W(\gamma) \to e^{i\pi} W(\gamma) \,. \tag{3.42}$$

Conversely, consider a local region around $v_0$ and suppose the magnetic defect $M$ comes from the right and ends on the dual vertex associated with the 2-simplex $[v_0, v_1, v_2]$. That means that we should sum over gauge field configurations with $g_{(v_0, v_1, v_2)} = (0,1) \in \mathbb{Z}_2 \times \mathbb{Z}_2$. However, with no other insertions, in the presence of the topological action this configuration is not gauge invariant. Under the gauge transformation with $\vec{h}(v_0) = (1,0)$ considered before the total action in this local region will changes to:

$$\alpha(0,0,0,1)\alpha(0,0,0,0)^2 = 1 \to \alpha(1,0,0,1)\alpha(1,0,0,0)^2 = e^{i\pi} \,, \tag{3.43}$$

where we used the explicit form for the algebraic cocycle (3.31). See Fig. 18 for illustration.

We see that a Wilson line and a magnetic defect cannot end on the domain wall in isolation. However, if the Wilson line ends at a vertex of a simplex dual to the locus of a magnetic defect (with the framing specified by Figure 18), their variation under gauge transformations cancels, and the configuration becomes gauge invariant. Using the notation of (3.35) we thus derive:

$$\mathcal{D}_{\mathbb{Z}_2 \times \mathbb{Z}_2, \alpha_2} \cdot W = M \,, \qquad \mathcal{D}_{\mathbb{Z}_2 \times \mathbb{Z}_2, \alpha_2} \cdot M = W \,, \tag{3.44}$$

confirming that $\mathcal{D}_{\mathbb{Z}_2 \times \mathbb{Z}_2, \alpha_2}$ is the electric-magnetic duality symmetry defect.

As already mentioned in Section 3.2.4, this feature, illustrated for $G = \mathbb{Z}_2$ in $D = 3$ is a generic feature of domain walls with the form $\mathcal{D}_{G \times G, \alpha}$ and $\alpha \in H^{D-1}(G \times G, U(1))$ non-factorized. Therefore, they generalize electric-magnetic duality to non-abelian groups and to higher dimensions. In higher dimensions, there are gauge invariant configurations with the codimension-two magnetic defect and the codimension-two invertible electric operators defined in (2.13) ending on the same codimension-three locus of the domain wall $\mathcal{D}_{G \times G, \alpha}$. For $D = 3$ the invertible electric operators of codimension-two are the invertible Wilson lines.

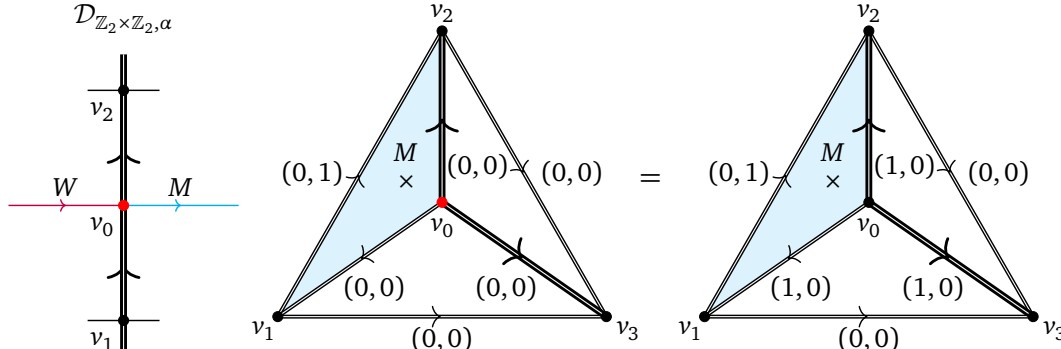

Figure 18: The first figure shows a configuration with a magnetic defect $M$ and a Wilson line $W$ ending on the same locus of the domain wall $\mathcal{D}_{\mathbb{Z}_2 \times \mathbb{Z}_2, \alpha_2}$. The other two figures shows a local slice of $\mathcal{D}_{\mathbb{Z}_2 \times \mathbb{Z}_2, \alpha_2}$ around this locus. Because the magnetic defect $M$ ends on the dual vertex associated with the 2-simplex $[v_0, v_1, v_2]$, we have $g_{(v_0, v_1, v_2)} = (0, 1)$. The presence of the magnetic defect makes the total action to changes under gauge transformations with $\vec{h}(v_0) = (1, 0) \in \mathbb{Z}_2 \times \mathbb{Z}_2$. This change, however, is compensated by the change of the open Wilson line ending on $v_0$ coming from the left. We see that the mechanism determining the action is reminiscent of anomaly inflow [83].

## 3.4 Higher codimensional topological operators: Cheshire strings

In the definition of the domain wall $\mathcal{D}_{H,\alpha}(\Sigma)$ with $H < G \times G$ and $\alpha \in H^{D-1}(H, U(1))$, a crucial ingredient is the orientation of the normal bundle $N\Sigma$, which gives a consistent global meaning to the "left" and "right" of the domain wall. Note, however, that diagonal domain walls can be constructed without this structure, i.e., diagonal domain walls are orientation-reversal invariant (see Section 3.1). Therefore, it is possible to generalize them to higher codimensional operators. From an alternative perspective, note that the holonomy of a contractible path crossing $\Sigma$ is trivial for a diagonal domain wall (see Fig. 2). This feature suggests that these domain walls can be constructed as condensations of Wilson lines [100]. Consequently, it is expected that such domain walls can be generalized to higher codimensional operators through this construction.

The $n$-dimensional generalization of diagonal domain walls is classified by:

- Subgroup $K \leq G$;

- Topological action $\alpha_n \in H^n(K, U(1))$.

Given the data $(K, \alpha_n)$ we define the $n$-dimensional operator $\mathcal{D}_{K,\alpha_n}(\Sigma_n)$ by restricting the gauge group elements and gauge transformations along the $n$-dimensional submanifold $\Sigma_n$ to be elements of $K$ and by decorating $\Sigma_n$ with the topological action $\alpha_n \in H^n(K, U(1))$ as in (2.13). In $D = 3 + 1$ and $n = 2$ these surface defects are discussed in [64].

The fusion rule of higher codimensional diagonal defects can be computed similarly to (3.7) and gives:

$$\mathcal{D}_{K,\alpha_n}(\Sigma_n) \times \mathcal{D}_{K',\alpha_n'}(\Sigma_n) = \frac{|G|}{|K \cdot K'|} \mathcal{D}_{K \cap K', \alpha_n \cdot \alpha_n'}(\Sigma_n). \tag{3.45}$$

For $G = \mathbb{Z}_2$, and trivial $\alpha_n$, the fusion (3.45) gives $\mathcal{D}_1 \times \mathcal{D}_1 = 2\mathcal{D}_1$ which is the fusion rule of Cheshire strings [64, 65]. Note that this fusion rule is universal with respect to spacetime and operator dimension.

We remark that to generalize the domain walls that are not diagonal one would need to introduce a foliation structure, which is ubiquitous in fracton models (see e.g., [84–88]).

Table 3: Fusion table for domain walls of $G = \mathbb{Z}_N$ gauge theory with $N$ prime. Above, $\mathbb{Z}_N^{(m)} = \{(mn, n) : n \in \mathbb{Z}_N\} \lhd \mathbb{Z}_N \times \mathbb{Z}_N$ and $m \in \mathbb{Z}_N^\times \cong \mathrm{Aut}(\mathbb{Z}_N)$. Note that the fusion coefficients are decoupled topological quantum field theories on $\Sigma$, in particular, $|\mathbb{Z}_N| = \mathcal{Z}^0(\mathbb{Z}_N, \Sigma)$ (see (2.9) and recall that we define domain walls on connected submanifolds).

| Domain walls | $\mathcal{D}_{\mathbb{Z}_N^{(m')}}$ | $\mathcal{D}_1$ | $\mathcal{D}_{\mathbb{Z}_N \times \mathbb{Z}_N}$ | $\mathcal{D}_{1 \times \mathbb{Z}_N}$ | $\mathcal{D}_{\mathbb{Z}_N \times 1}$ |
|---|---|---|---|---|---|
| $\mathcal{D}_{\mathbb{Z}_N^{(m)}}$ | $\mathcal{D}_{\mathbb{Z}_N^{(mm')}}$ | $\mathcal{D}_1$ | $\mathcal{D}_{\mathbb{Z}_N \times \mathbb{Z}_N}$ | $\mathcal{D}_{1 \times \mathbb{Z}_N}$ | $\mathcal{D}_{\mathbb{Z}_N \times 1}$ |
| $\mathcal{D}_1$ | $\mathcal{D}_1$ | $|\mathbb{Z}_N|\mathcal{D}_1$ | $\mathcal{D}_{1 \times \mathbb{Z}_N}$ | $|\mathbb{Z}_N|\mathcal{D}_{1 \times \mathbb{Z}_N}$ | $\mathcal{D}_1$ |
| $\mathcal{D}_{\mathbb{Z}_N \times \mathbb{Z}_N}$ | $\mathcal{D}_{\mathbb{Z}_N \times \mathbb{Z}_N}$ | $\mathcal{D}_{\mathbb{Z}_N \times 1}$ | $\mathcal{Z}(\mathbb{Z}_N)\mathcal{D}_{\mathbb{Z}_N \times \mathbb{Z}_N}$ | $\mathcal{D}_{\mathbb{Z}_N \times \mathbb{Z}_N}$ | $\mathcal{Z}(\mathbb{Z}_N)\mathcal{D}_{\mathbb{Z}_N \times 1}$ |
| $\mathcal{D}_{1 \times \mathbb{Z}_N}$ | $\mathcal{D}_{1 \times \mathbb{Z}_N}$ | $\mathcal{D}_1$ | $\mathcal{Z}(\mathbb{Z}_N)\mathcal{D}_{1 \times \mathbb{Z}_N}$ | $\mathcal{D}_{1 \times \mathbb{Z}_N}$ | $\mathcal{Z}(\mathbb{Z}_N)\mathcal{D}_1$ |
| $\mathcal{D}_{\mathbb{Z}_N \times 1}$ | $\mathcal{D}_{\mathbb{Z}_N \times 1}$ | $|\mathbb{Z}_N|\mathcal{D}_{\mathbb{Z}_N \times 1}$ | $\mathcal{D}_{\mathbb{Z}_N \times \mathbb{Z}_N}$ | $|\mathbb{Z}_N|\mathcal{D}_{\mathbb{Z}_N \times \mathbb{Z}_N}$ | $\mathcal{D}_{\mathbb{Z}_N \times 1}$ |

# 4 Examples

In this section, we investigate two examples that have a Lagrangian description: $\mathbb{Z}_N$ gauge theory for $N$ prime in arbitrary spacetime dimensions and $\mathbb{D}_4$ gauge theory in $D = 3$. Using this description, we provide alternative derivations for the lattice results in these specific cases.

## 4.1  $\mathbb{Z}_N$ gauge theory

Consider $\mathbb{Z}_N$ gauge theory with $N$ prime in arbitrary spacetime dimensions. The domain walls that are universal with respect to dimension are: $\mathcal{D}_{\mathbb{Z}_N \times \mathbb{Z}_N}$, $\mathcal{D}_{1 \times \mathbb{Z}_N}$, $\mathcal{D}_{\mathbb{Z}_N \times 1}$, $\mathcal{D}_1$ and $\mathcal{D}_{\mathbb{Z}_N^{(m)}}$. The last one is the automorphism domain wall associated with the map that sends $n$ to $mn$ with $m \in \mathbb{Z}_N^\times$. Using the fusion rules derived in Section 3.2 we can write the "fusion table" for these domain walls which we present in 3.

The $\mathbb{Z}_N$ gauge theory has $N - 1$ non-trivial Wilson lines that can be generated by the representation that maps $1 \in \mathbb{Z}_N$ to $e^{\frac{2\pi i}{N}}$. We denote the generating Wilson line associated with this representation by $W$. Similarly, the theory has $N - 1$ non-trivial magnetic defects generated by $M$, which fixes the holonomy to be $1 \in \mathbb{Z}_N$. The transformation properties of these operators can be computed using the methods presented in Section 3.3 and are:

$$\mathcal{D}_1 \cdot M^k = 0\,, \qquad\qquad \mathcal{D}_{\mathbb{Z}_N \times \mathbb{Z}_N} \cdot W^k = 0\,, \tag{4.1}$$

$$\mathcal{D}_{\mathbb{Z}_N \times \mathbb{Z}_N} \cdot M^k = \sum_{n=0}^{N-1} M^n\,, \qquad\qquad \mathcal{D}_1 \cdot W^k = \sum_{n=0}^{N-1} W^n\,, \tag{4.2}$$

$$\mathcal{D}_{\mathbb{Z}_N^{(m)}} \cdot M^k = M^{mk}\,, \qquad\qquad \mathcal{D}_{\mathbb{Z}_N^{(m)}} \cdot W^k = W^{\frac{k}{m}}\,, \tag{4.3}$$

where we used the notation of (3.35) and $k \in \mathbb{Z}_N$ except in (4.1) where $k \in \mathbb{Z}_N^\times$, and the fact that $m$ is invertible. For the other factorized domain walls, the Wilson lines can end on the identity side, and magnetic defects on the $\mathbb{Z}_N$ side.

In addition, there are domain walls specific to each dimension obtained by attaching a topological action for the corresponding subgroup. For example, in $D = 3$ and $N = 2$, we have $\mathcal{D}_{\mathbb{Z}_2 \times \mathbb{Z}_2, \alpha_2}$ with $\alpha_2$ the non-trivial element of $H^2(\mathbb{Z}_2 \times \mathbb{Z}_2, U(1)) = \mathbb{Z}_2$. This domain wall squares to the identity as shown in (3.28) and corresponds to the electric-magnetic duality symmetry operator as shown in (3.44).

Table 4: Dictionary between gapped boundaries and boundary conditions in $\mathbb{Z}_N$ gauge theory.

| Gapped boundary | Boundary condition |
|:---:|:---:|
| $\mathcal{B}_1$ | $a = 0$ |
| $\mathcal{B}_{\mathbb{Z}_N}$ | $\tilde{a} = 0$ |

The $\mathbb{Z}_N$ gauge theory in arbitrary dimensions can be described by the BF action using two $U(1)$ gauge fields $a^{(1)}, \tilde{a}^{(D-2)}$ (the superscript denotes the form-degree):

$$S_{\text{BF}} = \frac{iN}{2\pi} \int_{\mathcal{M}} \tilde{a}^{(D-2)} \wedge da^{(1)}, \tag{4.4}$$

with gauge transformations $a^{(1)} \to a^{(1)} + d\alpha^{(0)}$ and $\tilde{a}^{(D-2)} \to \tilde{a}^{(D-2)} + d\tilde{\alpha}^{(D-3)}$. In this formulation, the generating Wilson line and magnetic defects are described by:

$$W(\gamma) = \exp\left(i \oint_\gamma a^{(1)}\right), \qquad M(\Gamma) = \exp\left(i \oint_\Gamma \tilde{a}^{(D-2)}\right), \tag{4.5}$$

with $\gamma$ a closed loop and $\Gamma$ a closed $(D-2)$-dimensional submanifold. We want to use this Lagrangian description to give an alternative derivation of the results presented above.

### 4.1.1 Gapped boundaries and boundary conditions

Consider the action (4.4) in a manifold $\mathcal{M}$ with boundary $\partial\mathcal{M}$. To define the theory it is necessary to choose boundary conditions. A boundary condition in this setup is a condition for the fields $a$ and $\tilde{a}$ such that there is no surface term in the variation of the action. The bulk equations of motion are standard and imply that all gauge fields are closed. Meanwhile, the boundary term in the variation of the action (4.4) is:

$$\delta S_{BF} = \frac{iN}{2\pi} \int_{\partial\mathcal{M}} \tilde{a}^{(D-2)} \wedge \delta a^{(1)}. \tag{4.6}$$

There are two boundary conditions for the fields $a$ and $\tilde{a}$ such that $\delta S_{BF} = 0$ which corresponds to the two subgroups of $\mathbb{Z}_N$ with $N$ prime (the trivial and the whole group). The dictionary is summarized in Table 4.

### 4.1.2 Domain walls and boundary conditions

To make contact with our lattice construction of domain walls we first divide spacetime into left and right regions $\mathcal{M} = \mathcal{M}_L \cup \mathcal{M}_R$ with a common boundary $\partial\mathcal{M}_L = -\partial\mathcal{M}_R = \Sigma$. The total action in this setup is

$$S_{\text{BF}} = \frac{iN}{2\pi} \int_{\mathcal{M}_L} \tilde{a}_L^{(D-2)} \wedge da_L^{(1)} + \frac{iN}{2\pi} \int_{\mathcal{M}_R} \tilde{a}_R^{(D-2)} \wedge da_R^{(1)}, \tag{4.7}$$

and one needs to specify boundary conditions on $\Sigma$. The bulk equations of motion are standard and imply that all gauge fields are closed. Meanwhile, the boundary term in the variation of the action (4.7) is:

$$\delta S_{\text{BF}} = \frac{iN}{2\pi} \int_{\Sigma} (\tilde{a}_L^{(D-2)} \wedge \delta a_L^{(1)} - \tilde{a}_R^{(D-2)} \wedge \delta a_R^{(1)}). \tag{4.8}$$

Table 5: Correspondence between domain walls and boundary conditions for $\mathbb{Z}_N$ gauge theory.

| Domain wall | Boundary condition |
|---|---|
| $\mathcal{D}_{\mathbb{Z}_N^{(m)}}$ | $a_L = m a_R$ and $m \tilde{a}_L = \tilde{a}_R$ |
| $\mathcal{D}_1$ | $a_L = a_R = 0$ |
| $\mathcal{D}_{\mathbb{Z}_N \times \mathbb{Z}_N}$ | $\tilde{a}_L = \tilde{a}_R = 0$ |
| $\mathcal{D}_{1 \times \mathbb{Z}_N}$ | $a_L = \tilde{a}_R = 0$ |
| $\mathcal{D}_{\mathbb{Z}_N \times 1}$ | $\tilde{a}_L = a_R = 0$ |

There is a correspondence between boundary conditions and domain walls that we summarize in Table 5. The boundary conditions in Table 5 give the exhaustive set of conditions for the fields such that $\delta S_{\mathrm{BF}} = 0$, and correspond to the domain walls that are universal with respect to dimension.

There are other classes of boundary conditions that arise by adding a boundary action term. Because these terms depend on the dimension, these boundary conditions are specific to each dimension and correspond to the twisted domain walls. For simplicity, consider $N = 2$, $D = 3$ and the boundary action:

$$S_{\alpha_2} = \frac{i}{\pi} \int_\Sigma a_L^{(1)} \wedge a_R^{(1)}. \tag{4.9}$$

With this additional boundary interaction term for $a_L^{(1)}$ and $a_R^{(1)}$, the variation of the total action becomes:

$$\delta(S_{\mathrm{BF}} + S_{\alpha_2}) = \frac{i}{\pi} \int_\Sigma \left[ (\tilde{a}_L^{(1)} - a_R^{(1)}) \wedge \delta a_L^{(1)} + (a_L^{(1)} - \tilde{a}_R^{(1)}) \wedge \delta a_R^{(1)} \right], \tag{4.10}$$

enabling the additional boundary condition with $\tilde{a}_L^{(1)} = a_R^{(1)}$ and $a_L^{(1)} = \tilde{a}_R^{(1)}$ on $\Sigma$. This boundary condition corresponds to the electric-magnetic duality domain wall $\mathcal{D}_{\mathbb{Z}_2 \times \mathbb{Z}_2, \alpha_2}$ where $\alpha_2$ is the non-trivial element of $H^2(\mathbb{Z}_2 \times \mathbb{Z}_2, U(1)) = \mathbb{Z}_2$.

More generally, the domain wall $\mathcal{D}_{\mathbb{Z}_N \times \mathbb{Z}_N, \alpha_{D-1}}$ associated with an element of $\alpha_{D-1} \in H^{D-1}(\mathbb{Z}_N \times \mathbb{Z}_N, U(1))$ corresponds to a boundary condition that is enabled by adding the corresponding boundary action term constructed from $a_L$ and $a_R$. The non-factorized terms, such as (4.9), give boundary conditions that relate the $a^{(1)}$ and $b^{(D-2)}$ fields on the two sides which implies that they transform electric into magnetic operators and generalize electric-magnetic duality. For example, in $D = 4$ we have $H^3(\mathbb{Z}_2 \times \mathbb{Z}_2, U(1)) = \mathbb{Z}_2 \times \mathbb{Z}_2 \times \mathbb{Z}_2$. The boundary condition that corresponds to the domain wall associated with the non-factorized element of this cohomology group is $b_L^{(2)} = d a_R^{(1)}$ and $d a_L^{(1)} = b_R^{(2)}$, which is enabled by the boundary action:

$$S_{\alpha_3} = \frac{i}{2\pi} \int_\Sigma a_L^{(1)} \wedge d a_R^{(1)}. \tag{4.11}$$

As we are going to show, this domain wall is non-invertible. The other two generators of $H^3(\mathbb{Z}_2 \times \mathbb{Z}_2, U(1))$ correspond to the boundary conditions $b_L^{(2)} = d a_L^{(1)}$ and $d a_R^{(1)} = b_R^{(2)}$, which are enabled by the boundary actions with $a_L \wedge d a_L$ and $a_R \wedge d a_R$ respectively. These other choices do not permute electric and magnetic objects, instead, the magnetic object decorated with the invertible electric operator 2.13 can end on it.

### 4.1.3 Fusion rules

Here we provide an alternative derivation of the fusion rules (3.3), (3.7) and (3.9) for the $G = \mathbb{Z}_N$ case. Then, we rederive (3.28) and its generalization to $D = 4$.

**Automorphism domain wall**   The domain wall $\mathcal{D}_{\mathbb{Z}_N^{(m)}}(\Sigma)$ corresponds to the boundary condition:

$$\mathcal{D}_{\mathbb{Z}_N^{(m)}}(\Sigma): \qquad a_L^{(1)}\Big|_\Sigma = m a_R^{(1)}\Big|_\Sigma, \qquad m \tilde{a}_L^{(D-2)}\Big|_\Sigma = \tilde{a}_R^{(D-2)}\Big|_\Sigma. \tag{4.12}$$

To compute the fusion rule $\mathcal{D}_{\mathbb{Z}_N^{(m)}} \times \mathcal{D}_{\mathbb{Z}_N^{(m')}}$ we can bring two such parallel surfaces on top of each other. We denote the gauge fields on the left, middle, and right layer by $a_L, \tilde{a}_L, a_M, \tilde{a}_M$, and $a_R, \tilde{a}_R$ respectively. Somewhat trivially, the boundary conditions combine: $a_L = m a_M$ & $a_M = m' a_R \Rightarrow a_L = mm' a_R$ and $m \tilde{a}_L = \tilde{a}_M$ & $m' \tilde{a}_M = \tilde{a}_R \Rightarrow mm' \tilde{a}_L = \tilde{a}_R$ so that

$$\mathcal{D}_{\mathbb{Z}_N^{(m)}}(\Sigma) \times \mathcal{D}_{\mathbb{Z}_N^{(m')}}(\Sigma): \qquad a_L^{(1)}\Big|_\Sigma = mm' a_R^{(1)}\Big|_\Sigma, \quad mm' \tilde{a}_L^{(D-2)}\Big|_\Sigma = \tilde{a}_R^{(D-2)}\Big|_\Sigma, \tag{4.13}$$

which is the boundary condition for $\mathcal{D}_{\mathbb{Z}_N^{(mm')}}$. Note that there is no remnant degrees of freedom to sum over, this is in contrast with the next two cases. We conclude that

$$\mathcal{D}_{\mathbb{Z}_N^{(m)}} \times \mathcal{D}_{\mathbb{Z}_N^{(m')}} = \mathcal{D}_{\mathbb{Z}_N^{(mm')}}, \tag{4.14}$$

which is consistent with (3.3).

**Diagonal domain wall**   The domain wall $\mathcal{D}_1(\Sigma)$ corresponds to having Dirichlet boundary condition for $a_L$ and $a_R$ along $\Sigma$:

$$\mathcal{D}_1(\Sigma): \qquad a_L^{(1)}\Big|_\Sigma = a_R^{(1)}\Big|_\Sigma = 0. \tag{4.15}$$

This boundary condition results in trivial holonomy for loops on $\Sigma$. This is also the case for $\mathcal{D}_1(\Sigma)$ as illustrated in Fig. 2. From the method presented for the automorphism domain wall, it is clear that $\mathcal{D}_1 \times \mathcal{D}_1$ should be proportional to itself because $a_L = a_M = a_R$. The difference in this case, is that $\tilde{a}_M^{(D-2)}$ is a remant degree of freedom along $\Sigma$ that we should sum over. This gives the fusion coefficient $\mathcal{Z}^0(\mathbb{Z}_N, \Sigma) = |\mathbb{Z}_N|$.

To see this more explicitly, instead of defining the domain walls by imposing the boundary conditions, we are going to add scalar fields with suitable gauge transformations to cancel the variation of the total action. These fields are sometimes referred to as Stueckelberg scalars, see e.g., [7,89] and the mechanism is somewhat analogous to anomaly inflow. Here, we will generalize in dimension the presentation of section 6.3.4 of [90] which we refer to for further details.[5] The Dirichlet boundary condition for $a_L$ and $a_R$ can be implemented by the boundary action:

$$\mathcal{D}_1(\Sigma): \qquad -\frac{iN}{2\pi} \int_\Sigma \phi^{(0)} d(\tilde{a}_L^{(D-2)} - \tilde{a}_R^{(D-2)}), \tag{4.16}$$

with $\phi^{(0)}$ the Stueckelberg scalar field with gauge transformation $\phi^{(0)} \to \phi^{(0)} + \alpha^{(0)}$ with $\alpha^{(0)}$ the gauge transformation of the $a$ fields. One can check that the total action is gauge invariant and, by integrating out $\tilde{a}$, that $a$ is pure gauge on $\Sigma$.

---

[5]In their analysis, the bulk action has the form $a_L^{(1)} \wedge d\tilde{a}_L^{(D-2)}$ which is related to (4.7) by the boundary term $a_L^{(1)} \wedge \tilde{a}_L^{(D-2)}$ under integration by parts. So we assume implicitly here that we added this boundary terms to get to their convention.

To compute the fusion rule of $\mathcal{D}_1$ with itself we can bring two such parallel surfaces on top of each other. We denote the gauge fields on the left, middle, and right layer by $a_L, \tilde{a}_L, a_M, \tilde{a}_M$, and $a_R, \tilde{a}_R$ respectively, and we denote the two Stueckelberg scalar fields by $\phi_1$ and $\phi_2$. The worldsheet action for the fusion is

$$\mathcal{D}_1(\Sigma) \times \mathcal{D}_1(\Sigma): \qquad -\frac{iN}{2\pi} \int_\Sigma \left[ -\phi^{(0)} d\tilde{a}^{(D-2)} + \phi_2^{(0)} (d\tilde{a}_L^{(D-2)} - d\tilde{a}_R^{(d-2)}) \right], \qquad (4.17)$$

with $\phi^{(0)} = \phi_1^{(0)} - \phi_2^{(0)}$ and $\tilde{a}^{(D-2)} = \tilde{a}_M^{(D-2)} - \tilde{a}_L^{(D-2)}$. The first term is the 0-form partition function for a decoupled scalar, i.e., $\mathcal{Z}^0(\mathbb{Z}_N, \Sigma) = |\mathbb{Z}_N|$, and the second term is another copy of the surface defect $\mathcal{D}_1(\Sigma)$. We conclude that

$$\mathcal{D}_1 \times \mathcal{D}_1 = |\mathbb{Z}_N| \mathcal{D}_1, \qquad (4.18)$$

which is consistent with expression (3.7). We see that the prefactor, which we derived before by summing over the intermediate holonomies, can be interpreted, in the abelian case, as the partition function for a decoupled scalar.

**Factorized domain wall**   The domain wall $\mathcal{D}_{\mathbb{Z}_N \times \mathbb{Z}_N}(\Sigma)$ corresponds to having Dirichlet boundary condition for $\tilde{a}_L$ and $\tilde{a}_R$:

$$\mathcal{D}_{\mathbb{Z}_N \times \mathbb{Z}_N}(\Sigma): \qquad \tilde{a}_L^{(D-2)} \Big|_\Sigma = \tilde{a}_R^{(D-2)} \Big|_\Sigma = 0. \qquad (4.19)$$

The result of this boundary condition is that the components of $da_L$ and $da_R$ that are perpendicular to $\Sigma$ are not set to zero after integrating out $\tilde{a}_L$ and $\tilde{a}_R$, therefore, the holonomy for contractible paths that cross $\Sigma$ is not trivial. This is also the case for $\mathcal{D}_{\mathbb{Z}_N \times \mathbb{Z}_N}(\Sigma)$ as illustrated in Fig. 2. In this case, the combination of boundary conditions gives $\tilde{a}_L = \tilde{a}_M = \tilde{a}_R$ and it remains to sum over $a_M^{(1)}$ along $\Sigma$, which gives the fusion coefficient $\mathcal{Z}(\mathbb{Z}_N, \Sigma)$.

We can see this more explicitly by following the same derivation as outlined before. In this case, we don't add the boundary term $a_L^{(1)} \wedge \tilde{a}_L^{(D-2)}$ and the Dirichlet boundary condition for $\tilde{a}_L$ and $\tilde{a}_R$ is equivalent to the boundary action:

$$\mathcal{D}_{\mathbb{Z}_N \times \mathbb{Z}_N}(\Sigma): \qquad -\frac{iN}{2\pi} \int_\Sigma \phi^{(D-3)} d(a_L^{(1)} - a_R^{(1)}), \qquad (4.20)$$

with $\phi^{(D-3)}$ the Stueckelberg scalar field with gauge transformation $\phi^{(D-3)} \to \phi^{(D-3)} - \tilde{\alpha}^{(D-3)}$ with $\tilde{\alpha}^{(D-3)}$ the gauge transformation of the field $\tilde{a}^{(D-2)}$. One can check that the total action is gauge invariant and, by integrating out $a$, that $\tilde{a}$ is pure gauge on $\Sigma$. The fusion in this case leads to:

$$\mathcal{D}_{\mathbb{Z}_N \times \mathbb{Z}_N}(\Sigma) \times \mathcal{D}_{\mathbb{Z}_N \times \mathbb{Z}_N}(\Sigma): \qquad -\frac{iN}{2\pi} \int_\Sigma \left[ -\phi^{(D-3)} \wedge da^{(1)} + \phi_2^{(D-3)} \wedge (da_L^{(1)} - da_R^{(1)}) \right], \quad (4.21)$$

with $\phi^{(D-3)} = \phi_1^{(D-3)} - \phi_2^{(D-3)}$ and $a^{(1)} = a_M^{(1)} - a_L^{(1)}$. The first term is now the partition function $\mathcal{Z}(\mathbb{Z}_N, \Sigma)$, and the second term is another copy of the surface defect $\mathcal{D}_{\mathbb{Z}_N \times \mathbb{Z}_N}$. We conclude that

$$\mathcal{D}_{\mathbb{Z}_N \times \mathbb{Z}_N} \times \mathcal{D}_{\mathbb{Z}_N \times \mathbb{Z}_N} = \mathcal{Z}(\mathbb{Z}_N) \mathcal{D}_{\mathbb{Z}_N \times \mathbb{Z}_N}, \qquad (4.22)$$

which is consistent with (3.9).

**Electric-magnetic duality domain wall in $D = 3$**

The electromagnetic duality that exchanges the electric and magnetic particles (see e.g. [91,92]) corresponds to the domain wall $\mathcal{D}_{\mathbb{Z}_2 \times \mathbb{Z}_2, \alpha_2}(\Sigma)$ correspond to the boundary conditions:

$$\mathcal{D}_{\mathbb{Z}_2 \times \mathbb{Z}_2, \alpha_2}(\Sigma) : \qquad a_L^{(1)}\Big|_\Sigma = \tilde{a}_R^{(1)}\Big|_\Sigma, \quad \tilde{a}_L^{(1)}\Big|_\Sigma = a_R^{(1)}\Big|_\Sigma. \tag{4.23}$$

that are enabled by the boundary action (4.9). Similarly to the automorphism fusion we have:

$$\mathcal{D}_{\mathbb{Z}_N \times \mathbb{Z}_N, \alpha_2}(\Sigma) \times \mathcal{D}_{\mathbb{Z}_N \times \mathbb{Z}_N, \alpha_2}(\Sigma) : \qquad a_L^{(1)}\Big|_\Sigma = a_R^{(1)}\Big|_\Sigma, \quad \tilde{a}_L^{(1)}\Big|_\Sigma = \tilde{a}_R^{(1)}\Big|_\Sigma. \tag{4.24}$$

which is the boundary condition for $\mathcal{D}_{\mathbb{Z}_2^{\mathrm{id}}} = 1$. Importantly combining the boundary actions leads to:

$$S_{\alpha_2, L} + S_{\alpha_2, R} = \frac{i}{\pi} \int_\Sigma a_L^{(1)} \wedge a_M^{(1)} + \frac{i}{\pi} \int_\Sigma a_M^{(1)} \wedge a_R^{(1)}. \tag{4.25}$$

Integrating out $a_M$ simply reproduces the relation $a_L = a_R$. We conclude that:

$$\mathcal{D}_{\mathbb{Z}_2 \times \mathbb{Z}_2, \alpha_2} \times \mathcal{D}_{\mathbb{Z}_2 \times \mathbb{Z}_2, \alpha_2} = 1, \tag{4.26}$$

which is (3.28).

**Non-invertible electric-magnetic duality domain wall in $D = 4$**  Consider a generalization of the previous defect to $D = 4$ corresponding to the boundary condition:

$$\mathcal{D}_{\mathbb{Z}_2 \times \mathbb{Z}_2, \alpha_3}(\Sigma) : \qquad da_L^{(1)}\Big|_\Sigma = \tilde{a}_R^{(2)}\Big|_\Sigma, \qquad \tilde{a}_L^{(2)}\Big|_\Sigma = da_R^{(1)}\Big|_\Sigma, \tag{4.27}$$

that is enabled by adding the boundary action (4.11). Now, in the fusion we get $da_L = da_R$ and $\tilde{a}_L^{(2)} = \tilde{a}_R^{(2)}$ which is not the identity boundary condition. The $a_L$ and $a_R$ gauge fields on the two sides after the fusion can differ by a $\mathbb{Z}_4$ gauge field, instead of being identically equal. The domain wall is non-invertible.

### 4.1.4 Transformation of Wilson lines and magnetic defects

One way to derive the correspondence between the boundary conditions and the domain walls of Table 5 is by the transformations of other operators. Conversely, given the correspondence, we can check the transformation of other operators using the boundary conditions.

**Untwisted domain walls**  Here, we derive the transformation property (3.37) for $G = \mathbb{Z}_N$ from the boundary condition $a_L = m a_R$ and $m \tilde{a}_L = \tilde{a}_R$. Consider a path $\gamma = \gamma_L + \gamma_R$ on $\mathcal{M}$ that cross $\Sigma$ with $\gamma_L$ in $\mathcal{M}_L$ and $\gamma_R$ on $\mathcal{M}_R$ such that $\partial \gamma_L = -\partial \gamma_R = v_1 - v_0$ with $v_1$ and $v_0$ vertices on $\Sigma$. Then, $W_L(\gamma_L) W_R^m(\gamma_R)$ is gauge invariant because

$$W_L(\gamma_L) W_R^m(\gamma_R) \rightarrow \exp\Big(\frac{i}{\pi} \int_{\gamma_L} a_L^{(1)} + d\alpha_L^{(0)} + \frac{im}{\pi} \int_{\gamma_R} \tilde{a}_R^{(1)} + d\alpha_R^{(0)}\Big) = W_L(\gamma_L) W_R^m(\gamma_R), \tag{4.28}$$

where we used

$$\int_{\gamma_L} d\alpha_L^{(0)} + m \int_{\gamma_R} d\alpha_R^{(0)} = \alpha_L^{(0)}(v_1) - m\alpha_R^{(0)}(v_0) + m\alpha_R^{(0)}(v_1) - \alpha_L^{(0)}(v_1) = 0, \tag{4.29}$$

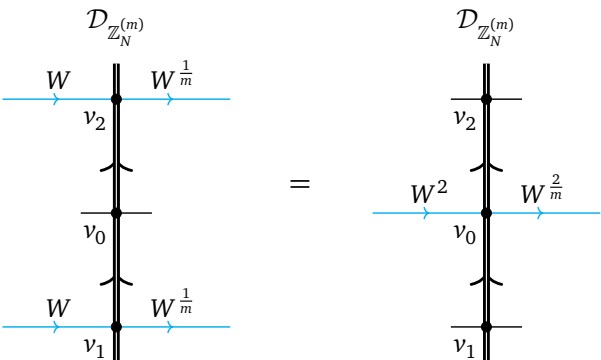

Figure 19: Derivation of the transformation of $W^2$ on $\mathcal{D}_{\mathbb{Z}_N^{(m)}}$ from the transformation of $W$. By moving the two independent configurations we can fuse them independently on each side.

which follows from the boundary condition $\alpha_L^{(0)}|_\Sigma = m\alpha_R^{(0)}|_\Sigma$. We conclude that

$$\mathcal{D}_{\mathbb{Z}_N^{(m)}} \cdot W = W^{\frac{1}{m}}, \tag{4.30}$$

which is consistent with (3.37). The transformation of the other Wilson lines and magnetic defects can be obtained similarly. Complementarily, they can also be obtained by fusion, see Fig. 19 for illustration.

The fact that Wilson lines and magnetic defects can end on the side with Dirichlet boundary conditions for $a$ and $\tilde{a}$ respectively is even easier to derive. For example, suppose we have $a_L^{(1)}|_\Sigma = 0$ and consider a path $\gamma_L$ such that $\partial \gamma_L = v_1 - v_0$ with $v_1$ and $v_0$ vertices on $\Sigma$. Then, under a gauge transformation we will have

$$W(\gamma_L) \rightarrow \exp\left(\frac{i}{\pi}\int_{\gamma_L} d\alpha_L^{(0)}\right) W(\gamma_L) = \exp\left(\alpha_L^{(0)}(v_1) - \alpha_L^{(0)}(v_0)\right) W(\gamma_L) = W(\gamma_L), \tag{4.31}$$

where we used the boundary condition $\alpha_L^{(0)}|_\Sigma = 0$. The same argument works for $a_R^{(1)}|_\Sigma = 0$ and for magnetic defects on the domain walls with Dirichlet boundary conditions for $\tilde{a}^{(D-2)}$. These results are consistent with the transformation properties of Wilson lines and magnetic defects for the other untwisted domain walls.

**Twisted domain walls**    Similarly to the automorphism domain wall derivation, the boundary conditions $\tilde{a}_L^{(1)} = a_R^{(1)}$ and $a_L^{(1)} = \tilde{a}_R^{(1)}$ implies:

$$\mathcal{D}_{\mathbb{Z}_2 \times \mathbb{Z}_2, \alpha_2} \cdot M = W, \qquad \mathcal{D}_{\mathbb{Z}_2 \times \mathbb{Z}_2, \alpha_2} \cdot W = M. \tag{4.32}$$

There is, however, a complementary point of view to understand this result which uses the analysis of twisted $\mathbb{Z}_2 \times \mathbb{Z}_2$ gauge theory in $D = 2$ from [7, 93]. This theory can be described by the action:

$$S_{\mathbb{Z}_2 \times \mathbb{Z}_2, \alpha_2} = \frac{i}{\pi}\int_\Sigma (\tilde{a}_L^{(0)} \wedge da_L^{(1)} + \tilde{a}_R^{(0)} \wedge da_R^{(1)} + a_L^{(1)} \wedge a_R^{(1)}), \tag{4.33}$$

where $\tilde{a}_L$ and $\tilde{a}_R$ are $2\pi$-periodic scalars and the system has gauge symmetry:

$$a_L^{(1)} \rightarrow a_L^{(1)} + d\alpha_L^{(0)}, \quad a_R^{(1)} \rightarrow a_R^{(1)} + d\alpha_R^{(0)}, \quad b_L^{(0)} \rightarrow b_L^{(0)} - \alpha_R^{(0)}, \quad b_R^{(0)} \rightarrow b_R^{(0)} - \alpha_L^{(0)}. \tag{4.34}$$

In this theory the local magnetic defects $M_L(v_i) = e^{i\tilde{a}_L(v_i)}$ and $M_R(v_i) = e^{i\tilde{a}_R(v_i)}$ are not gauge invariant. Instead, the gauge-invariant operators are $M_L(v_i)W_R(\gamma)M_L(v_j)$, and $M_R(v_i)W_L(\gamma)M_R(v_j)$ with $\gamma$ an open path with endpoints $v_i$ and $v_j$. If we extend $\gamma$ to the bulk and view $M_L(v_i)$ and $M_R(v_j)$ as the endpoints of a bulk magnetic operator, we again confirm the transformation (4.32).

## 4.2  $\mathbb{D}_4$ gauge theory

Consider the gauge theory for the Dihedral group of order 8 in $D = 3$. This theory is equivalent to twisted $\mathbb{Z}_2 \times \mathbb{Z}_2 \times \mathbb{Z}_2$ gauge theory and can be described by the action:

$$S_{\mathbb{D}_4} = \frac{i}{\pi} \int_{\mathcal{M}} \left( a \wedge d\tilde{a} + b \wedge d\tilde{b} + c \wedge d\tilde{c} + \frac{1}{\pi} a \wedge \tilde{b} \wedge c \right), \tag{4.35}$$

with $a, \tilde{a}, b, \tilde{b}, c, \tilde{c}$ one-form $U(1)$ gauge fields with correlated gauge transformations such that the action is gauge invariant on closed manifolds. Integrating out $\tilde{b}$ forces $\frac{1}{\pi} a \wedge c = db$, which describes the extension of $\mathbb{Z}_2 \times \mathbb{Z}_2$ by $\mathbb{Z}_2$ with the 2-cocycle given by $a \wedge b$. The equivalence with $\mathbb{D}_4$ gauge theory is discussed in [94–96].

As in the other examples, we divide spacetime into left and right regions $\mathcal{M} = \mathcal{M}_L \cup \mathcal{M}_R$ with a common boundary $\partial \mathcal{M}_L = -\partial \mathcal{M}_R = \Sigma$ with opposite orientation. The total action is (4.35) with fields defined on a left and right part, which we denote with the subscripts $L$ and $R$ as we have done in (4.7). In general, different boundary conditions correspond to different domain walls.

### 4.2.1  Diagonal fusion rule

Here we provide an alternative derivation of the fusion rule (3.7) for the $G = \mathbb{D}_4$ case. The domain wall $\mathcal{D}_1(\Sigma)$ corresponds to having Dirichlet boundary condition for $a, b, c$ along $\Sigma$:

$$\mathcal{D}_1(\Sigma): \qquad a_L\big|_\Sigma = a_R\big|_\Sigma = b_L\big|_\Sigma = b_R\big|_\Sigma = c_L\big|_\Sigma = c_R\big|_\Sigma = 0. \tag{4.36}$$

The result of this boundary condition is that the holonomy for loops on $\Sigma$ are always trivial which is also the case for $\mathcal{D}_1(\Sigma)$ as illustrated in Fig. 2. In complete analogy with the $\mathbb{Z}_N$ example, this boundary condition can be implemented by having Stueckeelberg scalar fields $\phi_a$, $\phi_b$ and $\phi_c$ along $\Sigma$. Similar manipulations, then yield three decoupled scalars that make a prefactor equal to $2^3 = |\mathbb{D}_4|$. We see that in this non-abelian example, the pre-factor can also be interpreted as the partition function for decoupled scalars.

### 4.2.2  Non-invertible electric-magnetic duality

The possible topological actions for the subgroup $H = \mathbb{D}_4 \times \mathbb{D}_4 \leq \mathbb{D}_4 \times \mathbb{D}_4$ are classified by $H^2(\mathbb{D}_4 \times \mathbb{D}_4, U(1)) = \mathbb{Z}_2 \times \mathbb{Z}_2$ (see [97] for the group cohomology of Dihedral groups). In the theory with $\tilde{b}$ integrated out, the domain wall $\mathcal{D}_{\mathbb{D}_4 \times \mathbb{D}_4,(n,m)}(\Sigma)$ (here $(n,m) \in \mathbb{Z}_2 \times \mathbb{Z}_2$) corresponds to the boundary condition:

$$\mathcal{D}_{\mathbb{D}_4 \times \mathbb{D}_4,(n,m)}(\Sigma): \quad \tilde{a}_L\big|_\Sigma = nc_R\big|_\Sigma, \quad na_L\big|_\Sigma = \tilde{c}_R\big|_\Sigma, \quad \tilde{a}_R\big|_\Sigma = mc_L\big|_\Sigma, \quad ma_R\big|_\Sigma = \tilde{c}_L\big|_\Sigma. \tag{4.37}$$

which is enabled by adding the boundary action:

$$S_{(n,m)} = \frac{in}{\pi} \int_\Sigma a_L \wedge c_R + \frac{im}{\pi} \int_\Sigma c_R \wedge b_L. \tag{4.38}$$

This correspondence means that the boundary condition in (4.37) solves $\delta(S_{\mathbb{D}_4} + S_{(n,m)}) = 0$, where $S_{\mathbb{D}_4}$ is (4.35) with spacetime divided into left and right regions as in (4.7). Note that in the theory with $\tilde{b}_L$ and $\tilde{b}_R$ integrated out $a_L \wedge c_L$ and $a_R \wedge c_R$ are exact, but not $a_L \wedge c_R$ and $a_R \wedge c_L$, which is what appears in (4.38).

**Fusion rules**  To compute the fusion rule of $\mathcal{D}_{\mathbb{D}_4 \times \mathbb{D}_4, (n,m)}$, instead of imposing the boundary conditions, we will add Stueckelberg fields to make the total action gauge invariant. Let us compute the fusion rule of two such domain walls by dividing the spacetime into the left, middle, and right. The boundary action (4.38) combines into $S_{(n_1,m_1),L} + S_{(n_1,m_1),R}$:

$$\frac{n_1 i}{\pi} \int a_L \wedge c_M + \frac{m_1 i}{\pi} \int a_M \wedge c_L + \frac{n_2 i}{\pi} \int a_M \wedge c_R + \frac{m_2 i}{\pi} \int a_R \wedge c_M. \tag{4.39}$$

Additionally, integrating out the Lagrange multiplier field associated with the condition that $a_M \wedge c_M$ is exact on $\Sigma$ generates:

$$1 + \exp\left( \frac{i}{\pi} \int_\Sigma a_M \wedge c_M \right). \tag{4.40}$$

The fusion outcome is different for the two terms in (4.40). For the first term, integrating out $a_M$ and $c_M$ leads to $n_1 a_L = m_2 a_R$ and $m_1 c_L = n_2 c_R$. For the second term, integrating out $a_M$ and $c_M$ leads to

$$\frac{i}{\pi} \int (n_1 a_L + m_2 a_R) \wedge (m_1 c_L + n_2 c_R) = \frac{n_1 n_2 i}{\pi} \int a_L \wedge c_R + \frac{m_1 m_2 i}{\pi} \int a_R \wedge c_L, \tag{4.41}$$

where the equality used $a_L \wedge c_L$ and $a_R \wedge c_R$ being exact. The above corresponds to the domain wall $\mathcal{D}_{\mathbb{D}_4 \times \mathbb{D}_4, (n_1 n_2, m_1 m_2)}(\Sigma)$.

We conclude that the domain wall is non-invertible and we have, for example:

$$\mathcal{D}_{\mathbb{D}_4 \times \mathbb{D}_4, (1,1)} \times \mathcal{D}_{\mathbb{D}_4 \times \mathbb{D}_4, (1,1)} = 1 + \mathcal{D}_{\mathbb{D}_4 \times \mathbb{D}_4, (1,1)}. \tag{4.42}$$

This fusion rule is strikingly similar to that of the Fibonacci anyons in $D = 3$ TQFTs [98]. The key difference is that our domain wall is higher dimensional.

**Transformation of other operators**  The theory has four non-trivial Wilson lines, four non-trivial magnetic defects, and thirteen non-trivial Dyons. With the trivial line, this makes up a total of 22 operators [99]. Among them, we have:

$$W_a(\gamma) = e^{i\int_\gamma a}, \qquad W_c(\gamma) = e^{i\int_\gamma c}, \qquad M_a(\gamma) = e^{i\int_\gamma \tilde{a}}, \qquad M_c(\gamma) = e^{i\int_\gamma \tilde{c}}, \tag{4.43}$$

where we used a local polarization to write the magnetic defects without a bounding surface [34]. From the boundary conditions (4.37) associated with the domain wall $\mathcal{D}_{\mathbb{D}_4 \times \mathbb{D}_4, (1,1)}(\Sigma)$, we can follow the procedure used in the derivation of (4.30) to find:

$$\mathcal{D}_{\mathbb{D}_4 \times \mathbb{D}_4, (1,1)} \cdot M_a = W_c + \dots \qquad \mathcal{D}_{\mathbb{D}_4 \times \mathbb{D}_4, (1,1)} \cdot W_c = M_a + \dots, \tag{4.44}$$

confirming that $\mathcal{D}_{\mathbb{D}_4 \times \mathbb{D}_4, (1,1)}$ is the symmetry defect that implements an electric-magnetic duality. From the two configurations displayed above, and provided the fusion rule $M_a \times W_c = M_a$ we can derive:

$$\mathcal{D}_{\mathbb{D}_4 \times \mathbb{D}_4, (1,1)} \cdot M_a = M_a + W_c, \tag{4.45}$$

which is consistent with the Fibonacci fusion rule 4.42. See Fig. 20 for an illustration.

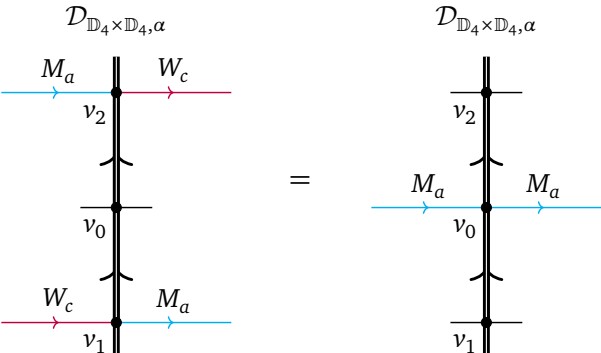

Figure 20: Derivation of (4.45) from the gauge invariant configurations (4.44) and the fusion rule $M_a \times W_c = M_a$. In the figure, we fused $M_a$ and $W_c$ on both sides. The fact that there exists a gauge invariant configuration with $M_a$ makes the transformation consistent with the Fibonacci fusion rule (4.42).

# Acknowledgments

We thank Maissam Barkeshli, Xie Chen, Arpit Dua, Ryohei Kobayashi, Michael Levin, Zhu-Xi Luo, Wilbur Shirley, and Carolyn Zhang for discussions. P.-S. H. thanks Kavli Institute for Theoretical Physics for hosting the program "Correlated Gapless Quantum Matter" in 2024, Perimeter Institute for hosting the conference "Physics of Quantum Information" in 2024, and University of Edinburgh for hosting the workshop "Categorical symmetries in Quantum Field Theory Workshop" in 2024, during which part of the work is completed. The authors are ordered alphabetically.

**Funding information** D.B.C. and C.C. are supported by the US Department of Energy Early Career program 5-29073, the Sloan Foundation, and the Simons Collaboration on Global Categorical Symmetries. P.-S.H. is supported by Simons Collaboration of Global Categorical Symmetry, Department of Mathematics King's College London, and also supported in part by grant NSF PHY-2309135 to the Kavli Institute for Theoretical Physics (KITP).

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
