# Peer review of "Non-invertible symmetries in finite group gauge theory"

_SciPost Physics, doi:SciPost Phys. 18, 019 (2025)_

## Round 2 · Referee Report · Anonymous (Referee 1) · 2024-8-2

Strengths

  1. Given the recent interest in non-invertible symmetries in the community, this work provides a timely addition to the literature.

  2. The paper is written in a pedagogical way and can be easily followed.

Weaknesses

  1. the result is systematic but not very surprising

  2. the authors set some limitations on the defects they consider. For example, they restricted to untwisted gauge theories and considered a subclass of their defects. It would be nice if the authors can comment on why they made the restriction or what would be the main difficulty to extend the result in this paper to the more general case.

Report

This paper studies the co-dimension one defects in untwisted gauge theories with a finite gauge group. It presents a systematic analysis of possible defects, both invertible and non-invertible, and studied their fusion rule and action on topological excitations. An example of a higher codimensional defect -- the Cheshire string -- is also briefly discussed. The paper is carefully written and can be a good addition to the literature. I think the paper can be published as it is. I would appreciate if the authors can consider point 2 in the `weakness' list.

Recommendation

Publish (easily meets expectations and criteria for this Journal; among top 50%)

---

## Round 2 · Referee Report · Anonymous (Referee 2) · 2024-10-21

Strengths

  1. The manuscript is very well written and easy to follow. The comprehensive review of finite gauge theory on the lattice is much appreciated.

  2. The manuscript contains several interesting gapped domain walls, such as non-invertible EM duality with a Fibonacci-like fusion rule and generalized Cheshire strings.

Weaknesses

  1. While the paper systematically computes and lists the fusion rules of gapped domain walls and their corresponding actions on gapped boundaries, the discussion on physical conclusions one can draw using these results is too brief.

Report

The authors provide a systematic presentation of the gapped domain walls in finite G gauge theories using the folding trick. They compute the fusion rules and the algebra of these gapped domain walls, as well as their actions on gapped boundary conditions. The results are easy to follow and presented pedagogically. The manuscript, as it stands, satisfies the journal's criteria and can be published.

My only suggestion is that it would be beneficial for the authors to add, if possible, an additional discussion on potential constraints when these gapped domain walls become symmetries. For instance, is the non-invertible EM duality anomalous?

I believe there is also a small typo on page 14. Using the convention in Eqs. (2.1) and (2.2), shouldn't the holonomy along the example path be $g_L g^{-1}_R h_R g_R g^{-1}_L h^{-1}_L$?

Recommendation

Publish (meets expectations and criteria for this Journal)

---

## Editorial Decision

published